# Continuous Wasserstein-2 Barycenter Estimation without Minimax Optimization

**Alexander Korotin**
Skolkovo Institute of Science and Technology
*Advanced Data Analytics in Science and Engineering Group*
Moscow, Russia
`a.korotin@skoltech.ru`

**Lingxiao Li**
Massachusetts Institute of Technology
*Geometric Data Processing Group*
Cambridge, Massachusetts, USA
`lingxiao@mit.edu`

**Justin Solomon**
Massachusetts Institute of Technology
*Geometric Data Processing Group*
Cambridge, Massachusetts, USA
`jsolomon@mit.edu`

**Evgeny Burnaev**
Skolkovo Institute of Science and Technology
*Advanced Data Analytics in Science and Engineering Group*
Moscow, Russia
`e.burnaev@skoltech.ru`

## ABSTRACT

Wasserstein barycenters provide a geometric notion of the weighted average of probability measures based on optimal transport. In this paper, we present a scalable algorithm to compute Wasserstein-2 barycenters given sample access to the input measures, which are not restricted to being discrete. While past approaches rely on entropic or quadratic regularization, we employ input convex neural networks and cycle-consistency regularization to avoid introducing bias. As a result, our approach does not resort to minimax optimization. We provide theoretical analysis on error bounds as well as empirical evidence of the effectiveness of the proposed approach in low-dimensional qualitative scenarios and high-dimensional quantitative experiments.

## 1 INTRODUCTION

Wasserstein barycenters have become popular due to their ability to represent the average of probability measures in a geometrically meaningful way. Techniques for computing Wasserstein barycenters have been successfully applied to many computational problems. In image processing, Wasserstein barycenters are used for color and style transfer (Rabin et al., 2014; Mroueh, 2019), and texture synthesis (Rabin et al., 2011). In geometry processing, shape interpolation can be done by computing barycenters (Solomon et al., 2015). In online machine learning, barycenters are used for aggregating probabilistic predictions of experts (Korotin et al., 2019b). Within the context of Bayesian inference, the barycenter of subset posteriors converges to the full data posterior, thus enabling efficient computational methods based on finding the barycenters (Srivastava et al., 2015; 2018).

Fast and accurate barycenter algorithms exist for discrete distributions (see Peyré et al. (2019) for a survey), while for continuous distributions the situation is more difficult and remains unexplored until recently (Li et al., 2020; Fan et al., 2020; Cohen et al., 2020). The discrete methods scale poorly with the number of support points of the barycenter and thus cannot approximate continuous barycenters well, especially in high dimensions.

In this paper, we present a method to compute Wasserstein-2 barycenters of *continuous* distributions based on a novel regularized dual formulation where the convex potentials are parameterized by input convex neural networks (Amos et al., 2017). Our algorithm is straightforward without introducing bias (e.g. Li et al. (2020)) or requiring minimax optimization (e.g. Fan et al. (2020)). This is made possible by combining a new *congruence regularizing term* combined with *cycle-consistency regularization* (Korotin et al., 2019a). As we will show in the analysis, thanks to the properties of

Wasserstein-2 distances, the gradients of the resulting convex potentials "push" the input distributions close to the true barycenter, allowing good approximation of the barycenter.

## 2 PRELIMINARIES

We denote the set of all Borel probability measures on $\mathbb{R}^D$ with finite second moment by $\mathcal{P}_2(\mathbb{R}^D)$. We use $\mathcal{P}_{2,ac}(\mathbb{R}^D) \subset \mathcal{P}_2(\mathbb{R}^D)$ to denote the subset of all absolutely continuous measures (w.r.t. the Lebesgue measure).

**Wasserstein-2 distance.** For $\mathbb{P}, \mathbb{Q} \in \mathcal{P}_2(\mathbb{R}^D)$, the **Wasserstein-2** distance is defined by

$$\mathbb{W}_2^2(\mathbb{P}, \mathbb{Q}) \stackrel{\text{def}}{=} \min_{\pi \in \Pi(\mathbb{P}, \mathbb{Q})} \int_{\mathbb{R}^D \times \mathbb{R}^D} \frac{\|x - y\|^2}{2} d\pi(x, y), \tag{1}$$

where $\Pi(\mathbb{P}, \mathbb{Q})$ is the set of probability measures on $\mathbb{R}^D \times \mathbb{R}^D$ whose marginals are $\mathbb{P}, \mathbb{Q}$, respectively. This definition is known as Kantorovich's primal form of transport distance (Kantorovitch, 1958).

The Wasserstein-2 distance $\mathbb{W}_2$ is well-studied in the theory of optimal transport (Brenier, 1991; McCann et al., 1995). In particular, it has a **dual formulation** (Villani, 2003):

$$\mathbb{W}_2^2(\mathbb{P}, \mathbb{Q}) = \int_{\mathbb{R}^D} \frac{\|x\|^2}{2} d\mathbb{P}(x) + \int_{\mathbb{R}^D} \frac{\|y\|^2}{2} d\mathbb{Q}(y) - \min_{\psi \in \text{Conv}} \left[ \int_{\mathbb{R}^D} \psi(x) d\mathbb{P}(x) + \int_{\mathbb{R}^D} \overline{\psi}(y) d\mathbb{Q}(y) \right], \tag{2}$$

where the minimum is taken over all the convex functions (potentials) $\psi : \mathbb{R}^D \to \mathbb{R} \cup \{\infty\}$, and $\overline{\psi}(y) = \max_{x \in \mathbb{R}^D} \left( \langle x, y \rangle - \psi(x) \right) : \mathbb{R}^D \to \mathbb{R} \cup \{\infty\}$ is the **convex conjugate** of $\psi$ (Fenchel, 1949), which is also a convex function. The optimal potential $\psi^*$ is defined up to an additive constant.

Brenier (1991) shows that if $\mathbb{P}$ does not give mass to sets of dimensions at most $D - 1$, then the optimal plan $\pi$ is uniquely determined by $\pi^* = [\text{id}_{\mathbb{R}^D}, T^*]\sharp\mathbb{P}$, where $T^* : \mathbb{R}^D \to \mathbb{R}^D$ is the unique solution to the Monge's problem

$$T^* = \arg\min_{T\sharp\mathbb{P}=\mathbb{Q}} \int_{\mathbb{R}^D} \frac{\|x - T(x)\|^2}{2} d\mathbb{P}(x). \tag{3}$$

The connection between $T^*$ and the dual formulation (2) is that $T^* = \nabla\psi^*$, where $\psi^*$ is the optimal solution of (2). Additionally, if $\mathbb{Q}$ does not give mass to sets of dimensions at most $D - 1$, then $T^*$ is invertible and

$$T^*(x) = \nabla\psi^*(x) = (\nabla\overline{\psi^*})^{-1}(x), \qquad (T^*)^{-1}(y) = \nabla\overline{\psi^*}(y) = (\nabla\psi^*)^{-1}(y).$$

In particular, the above discussion applies to the case where $\mathbb{P}, \mathbb{Q} \in \mathcal{P}_{2,ac}(\mathbb{R}^D)$.

**Wasserstein-2 barycenter.** Let $\mathbb{P}_1, \ldots, \mathbb{P}_N \in \mathcal{P}_{2,ac}(\mathbb{R}^D)$. Then, their barycenter w.r.t. weights $\alpha_1, \ldots, \alpha_N$ ($\alpha_n > 0$ and $\sum_{n=1}^N \alpha_n = 1$) is

$$\overline{\mathbb{P}} \stackrel{\text{def}}{=} \arg\min_{\mathbb{P} \in \mathcal{P}_2(\mathbb{R}^D)} \sum_{n=1}^N \alpha_n \mathbb{W}_2^2(\mathbb{P}_n, \mathbb{P}). \tag{4}$$

Throughout this paper, we assume that at least one of $\mathbb{P}_1, \ldots, \mathbb{P}_N \in \mathcal{P}_{2,ac}(\mathbb{R}^D)$ has bounded density. Under this assumption, $\overline{\mathbb{P}}$ is unique and absolutely continuous, i.e., $\overline{\mathbb{P}} \in \mathcal{P}_{2,ac}(\mathbb{R}^D)$, and it has bounded density (Agueh & Carlier, 2011, Definition 3.6 & Theorem 5.1).

For $n \in \{1, 2, \ldots, N\}$, let $(\psi_n^*, \overline{\psi_n^*})$ be the optimal pair of (mutually) conjugate potentials that transport $\mathbb{P}_n$ to $\overline{\mathbb{P}}$, i.e., $\nabla\psi_n^*\sharp\mathbb{P}_n = \overline{\mathbb{P}}$ and $\nabla\overline{\psi_n^*}\sharp\overline{\mathbb{P}} = \mathbb{P}_n$. Then $\{\overline{\psi_n^*}\}$ satisfy

$$\sum_{n=1}^N \alpha_n \nabla\overline{\psi_n^*}(x) = x \qquad \text{and} \qquad \sum_{n=1}^N \alpha_n \overline{\psi_n^*}(x) = \frac{\|x\|^2}{2} + c. \tag{5}$$

for all $x \in \mathbb{R}^D$ (Agueh & Carlier, 2011; Álvarez-Esteban et al., 2016). Since optimal potentials are defined up to a constant, for convenience, we set $c = 0$. The condition (5) serves as the basis for our algorithm for computing Wasserstein-2 barycenters. We say that potentials $\psi_1, \ldots, \psi_N$ are **congruent** w.r.t. weights $\alpha_1, \ldots, \alpha_n$ if their conjugate potentials satisfy (5), i.e., $\sum_{n=1}^D \alpha_n \overline{\psi_n}(x) = \frac{\|x\|^2}{2}$ for all $x \in \mathbb{R}^D$.

## 3 RELATED WORK

Most algorithms in the field of computational optimal transport are designed for the discrete setting where the input distributions have finite support; see the recent survey by Peyré et al. (2019) for discussion. A particular popular line of algorithms are based on entropic regularization that gives rise to the famous Sinkhorn iteration (Cuturi, 2013; Cuturi & Doucet, 2014). These methods are typically limited to a support of $10^5 - 10^6$ points before the problem becomes computationally infeasible. Similarly, discrete barycenter methods (Cuturi & Doucet, 2014), particularly the ones that rely on a fixed support for the barycenter (Dvurechenskii et al., 2018; Staib et al., 2017), cannot provide precise approximation of continuous barycenters in high dimensions, since a large number of samples is needed; see experiments in Fan et al. (2020, §4.3) for an example. Thus we focus on the existing literature in the continuous setting.

**Computation of Wasserstein-2 distances and maps.**    Genevay et al. (2016) demonstrate the possibility of computing Wasserstein distances given only sample access to the distributions by parameterizing the dual potentials as functions in the reproducing kernel Hilbert spaces. Based on this realization, Seguy et al. (2017) propose a similar method but use neural networks to parameterize the potentials, using entropic or $\mathcal{L}^2$ regularization w.r.t. $\mathbb{P} \times \mathbb{Q}$ to keep the potentials approximately conjugate. The transport map is recovered from optimized potentials via barycentric projection.

As we note in §2, $\mathbb{W}_2$ enjoys many useful theoretical properties. For example, the optimal potential $\psi^*$ is convex, and the corresponding optimal transport map is given by $\nabla \psi^*$. By exploiting these properties, Makkuva et al. (2019) propose a minimax optimization algorithm for recovering transport maps, using input convex neural networks (ICNNs) (Amos et al., 2017) to approximate the potentials.

An alternative to entropic regularization is the **cycle-consistency** regularization proposed by Korotin et al. (2019a). It uses the property that the gradients of optimal dual potentials are inverses of each other. The imposed regularizer requires integration only over the marginal measures $\mathbb{P}$ and $\mathbb{Q}$, instead of over $\mathbb{P} \times \mathbb{Q}$ as required by entropy-based alternatives. Their method converges faster than the minimax method since it does not have an inner optimization cycle.

Xie et al. (2019) propose using two generative models with a shared latent space to implicitly compute the optimal transport correspondence between $\mathbb{P}$ and $\mathbb{Q}$. Based on the obtained correspondence, the authors are able to compute the optimal transport distance between the distributions.

**Computation of Wasserstein-2 barycenters.**    A few recent techniques tackle the barycenter problem (4) using continuous rather than discrete approximations of the barycenter:

- MEASURE-BASED (GENERATIVE) OPTIMIZATION: Problem (4) optimizes over probability measures. This can be done using the generic algorithm by Cohen et al. (2020) who employ generative networks to compute barycenters w.r.t. arbitrary discrepancies. They test their method with the maximum mean discrepancy (MMD) and Sinkhorn divergence. This approach suffers from the usual limitations of generative models such as mode collapse. Applying it to $\mathbb{W}_2$ barycenters requires estimation of $\mathbb{W}_2^2(\mathbb{P}_n, \mathbb{P})$. Fan et al. (2020) test this approach using the minimax method by Makkuva et al. (2019), but they end up with a challenging *min-max-min* problem.
- POTENTIAL-BASED OPTIMIZATION: Li et al. (2020) recover the optimal potentials $\{\psi_n^*\}$ via a non-minimax regularized dual formulation. No generative model is needed: the barycenter is recovered by pushing forward measures using gradients of potentials or by barycentric projection.

## 4 METHODS

Inspired by Li et al. (2020) we use a potential-based approach and recover the barycenter by using gradients of the potentials as pushforward maps. The main differences are: (1) we restrict the potentials to be convex, (2) we enforce congruence via a regularizing term, and (3) our formulation does not introduce bias, meaning the optimal solution of our formulation gives the true barycenter.

### 4.1 DERIVING THE DUAL PROBLEM

Let $\overline{\mathbb{P}}$ be the true barycenter. Our goal is to recover the optimal potentials $\{\psi_n^*, \overline{\psi_n^*}\}$ mapping the input measures $\mathbb{P}_n$ into $\overline{\mathbb{P}}$.

To start, we express the barycenter objective (4) after substituting the dual formulation (2):

$$\sum_{n=1}^{N} \alpha_n \mathbb{W}_2^2(\mathbb{P}_n, \overline{\mathbb{P}}) = \left[ \sum_{n=1}^{N} \alpha_n \int_{\mathbb{R}^D} \frac{\|x\|^2}{2} d\mathbb{P}_n(x) \right] + \int_{\mathbb{R}^D} \frac{\|y\|^2}{2} d\overline{\mathbb{P}}(y) -$$

$$\min_{\{\psi_n\} \in \text{Conv}} \left[ \sum_{n=1}^{N} \alpha_n \int_{\mathbb{R}^D} \psi_n(x) d\mathbb{P}_n(x) + \sum_{n=1}^{N} \alpha_n \int_{\mathbb{R}^D} \overline{\psi_n}(y) d\overline{\mathbb{P}}(y) \right] \qquad (6)$$

The minimum is attained not just among convex potentials $\{\psi_n\}$, but among congruent potentials (see discussion under (5)); thus, we can add the constraint that $\{\psi_n\}$ are congruent to (6). Hence,

$$\sum_{n=1}^{N} \alpha_n \mathbb{W}_2^2(\mathbb{P}_n, \overline{\mathbb{P}}) = \left[ \sum_{n=1}^{N} \alpha_n \int_{\mathbb{R}^D} \frac{\|x\|^2}{2} d\mathbb{P}_n(x) \right] - \min_{\{\psi_n\} \text{ congruent}} \left[ \underbrace{\sum_{n=1}^{N} \alpha_n \int_{\mathbb{R}^D} \psi_n(y) d\mathbb{P}_n(y)}_{\text{MultiCorr}(\{\alpha_n, \mathbb{P}_n\}|\{\psi_n\})} \right]. \qquad (7)$$

To transition from (6) to (7), we used the fact that for congruent $\{\psi_n\}$ we have $\sum_{n=1}^{N} \alpha_n \overline{\psi_n}(x) = \frac{\|x\|^2}{2}$, so $\sum_{n=1}^{N} \int_{\mathbb{R}^D} \alpha_n \overline{\psi_n}(y) d\overline{\mathbb{P}}(y) = \int_{\mathbb{R}^D} \frac{\|y\|^2}{2} d\overline{\mathbb{P}}(y)$.

We call the value inside the minimum in (7) the **multiple correlation** of $\{\mathbb{P}_n\}$ with weights $\{\alpha_n\}$ w.r.t. potentials $\{\psi_n\}$. Notice that the true barycenter $\overline{\mathbb{P}}$ appears nowhere on the right side of (7). Thus the optimal potentials $\{\psi_n^*\}$ can be recovered by solving the following

$$\min_{\{\psi_n\} \text{ congruent}} \text{MultiCorr}(\{\alpha_n, \mathbb{P}_n\}|\{\psi_n\}) = \min_{\{\psi_n\} \text{ congruent}} \left[ \sum_{n=1}^{N} \alpha_n \int_{\mathbb{R}^D} \psi_n(y) d\mathbb{P}_n(y) \right]. \qquad (8)$$

### 4.2 IMPOSING THE CONGRUENCE CONDITION

It is challenging to impose the congruence condition on convex potentials. What if we relax the congruence condition? The following theorem bounds how close a set of convex potentials $\{\psi_n\}$ is to $\{\psi_n^*\}$ in terms of the difference of multiple correlation.

**Theorem 4.1.** *Let $\overline{\mathbb{P}} \in \mathcal{P}_{2,ac}(\mathbb{R}^D)$ be the barycenter of $\mathbb{P}_1, \ldots, \mathbb{P}_N \in \mathcal{P}_{2,ac}(\mathbb{R}^D)$ w.r.t. weights $\alpha_1, \ldots, \alpha_N$. Let $\{\psi_n^*\}$ be the optimal congruent potentials of the barycenter problem. Suppose we have $\mathcal{B}$-smooth[1] convex potentials $\{\psi_n\}$ for some $\mathcal{B} \in [0, +\infty]$, and denote $\Delta = \text{MultiCorr}(\{\alpha_n, \mathbb{P}_n\} \mid \{\psi_n\}) - \text{MultiCorr}(\{\alpha_n, \mathbb{P}_n\} \mid \{\psi_n^*\})$. Then,*

$$\Delta + \underbrace{\int_{\mathbb{R}^D} \sum_{n=1}^{N} \left[ \alpha_n \overline{\psi_n}(y) - \frac{\|y\|^2}{2} \right] d\overline{\mathbb{P}}(y)}_{\text{Congruence mismatch}} \geq \frac{1}{2\mathcal{B}} \sum_{n=1}^{N} \alpha_n \|\nabla \psi_n^*(x) - \nabla \psi_n(x)\|_{\mathbb{P}_n}^2. \qquad (9)$$

*Here $\|\cdot\|_\mu$ denotes the norm induced by inner product in Hilbert space $\mathcal{L}^2(\mathbb{R}^D \to \mathbb{R}^D, \mu)$. We call the second term on the left of (9) the **congruence mismatch**.*

We prove this in Appendix B. Note that if the congruence mismatch is non-positive, then

$$\Delta \geq \frac{1}{2\mathcal{B}} \sum_{n=1}^{N} \alpha_n \|\nabla \psi_n^*(x) - \nabla \psi_n(x)\|_{\mathbb{P}_n}^2 \geq \frac{1}{\mathcal{B}} \sum_{n=1}^{N} \alpha_n \mathbb{W}_2^2(\nabla \psi_n \sharp \mathbb{P}_n, \overline{\mathbb{P}}), \qquad (10)$$

where the last inequality of (10) follows from (Korotin et al., 2019a, Lemma A.2). From (10), we conclude that for all $n \in \{1, \ldots, N\}$, we have $\mathbb{W}_2^2(\nabla \psi_n \sharp \mathbb{P}_n, \overline{\mathbb{P}}) \leq \frac{\mathcal{B}\Delta}{\alpha_n}$. This shows that if the congruence mismatch is non-positive, then $\Delta$, the difference in multiple correlation, provides

---

[1]We say that a diffirentiable function $f : \mathbb{R}^D \to \mathbb{R}$ is $\mathcal{B}$-smooth if its gradient $\nabla f$ is $\mathcal{B}$-Lipschitz.

an upper bound for the Wasserstein-2 distance between the true barycenter and each pushforward $\nabla\psi_n \sharp \mathbb{P}_n$. This justifies the use of $\nabla\psi_n \sharp \mathbb{P}_n$ to recover the barycenter. Notice for optimal potentials, the congruence mismatch is zero.

Thus to penalize positive congruence mismatch, we introduce a regularizing term

$$\mathcal{R}_1^{\overline{\mathbb{P}}}(\{\alpha_n\}, \{\overline{\psi_n}\}) \stackrel{\text{def}}{=} \int_{\mathbb{R}^D} \left[\sum_{n=1}^N \alpha_n \overline{\psi_n}(y) - \frac{\|y\|^2}{2}\right]_+ d\overline{\mathbb{P}}(y). \tag{11}$$

Because we take the positive part of the integrand of (9) to get (11) and that the right side of (9) is non-negative, we have

$$\left[\text{MultiCorr}(\{\alpha_n, \mathbb{P}_n\} \mid \{\psi_n\}) + 1 \cdot \mathcal{R}_1^{\overline{\mathbb{P}}}(\{\alpha_n\}, \{\overline{\psi_n}\})\right] - \text{MultiCorr}(\{\alpha_n, \mathbb{P}_n\} \mid \{\psi_n^*\}) \geq 0$$

for all convex potentials $\{\psi_n\}$. On the other hand, for optimal potentials $\{\psi_n\} = \{\psi_n^*\}$, the inequality turns into equality, implying that adding the regularizing term $1 \cdot \mathcal{R}_1^{\overline{\mathbb{P}}}(\{\alpha_n\}, \{\overline{\psi_n}\})$ to (8) will not introduce bias – the optimal solution still yields $\{\psi_n^*\}$.

However, evaluating (11) exactly requires knowing the true barycenter $\overline{\mathbb{P}}$ a priori. To remedy this issue, one may replace $\overline{\mathbb{P}}$ with another absolutely continuous measure $\tau \cdot \widehat{\mathbb{P}}$ ($\tau \geq 1$ and $\widehat{\mathbb{P}}$ is a probability measure) whose density bounds that of $\overline{\mathbb{P}}$ from above almost everywhere. In this case,

$$\tau \cdot \mathcal{R}_1^{\widehat{\mathbb{P}}}(\{\alpha_n\}, \{\overline{\psi_n}\}) = \tau \cdot \int_{\mathbb{R}^D} \left[\sum_{n=1}^N \alpha_n \overline{\psi_n}(y) - \frac{\|y\|^2}{2}\right]_+ d\widehat{\mathbb{P}} \geq \mathcal{R}_1^{\overline{\mathbb{P}}}(\{\alpha_n\}, \{\overline{\psi_n}\}). \tag{12}$$

Hence we obtain the following regularized version of (8) where $\{\psi_n^*\}$ is the optimal solution:

$$\min_{\{\psi_n\} \in \text{Conv}} \left[\text{MultiCorr}(\{\alpha_n, \mathbb{P}_n\} \mid \{\psi_n\}) + \tau \cdot \mathcal{R}_1^{\widehat{\mathbb{P}}}(\{\alpha_n\}, \{\overline{\psi_n}\})\right]. \tag{13}$$

Selecting a measure $\tau \cdot \widehat{\mathbb{P}}$ is not obvious. Consider the case when $\{\mathbb{P}_n\}$ are supported on compact sets $\mathcal{X}_1, \ldots, \mathcal{X}_N \subset \mathbb{R}^D$ and $\mathbb{P}_1$ has density upper bounded by $h < \infty$. In this scenario, the barycenter density is upper bounded by $h \cdot \alpha_1^{-D}$ (Álvarez-Esteban et al., 2016, Remark 3.2). Thus, the measure $\tau \cdot \widehat{\mathbb{P}}$ supported on ConvexHull($\mathcal{X}_1, \ldots, \mathcal{X}_N$) with this density is an upper bound for $\overline{\mathbb{P}}$. We will address the question of how to choose $\tau, \widehat{\mathbb{P}}$ properly in practice in §4.4.

### 4.3 ENFORCING CONJUGACY OF POTENTIALS PAIRS

Throughout this subsection, we assume the upper bound finite measure $\tau \cdot \widehat{\mathbb{P}}$ of the $\overline{\mathbb{P}}$ is known. The optimization problem (13) involves not only the potentials $\{\psi_n\}$, but also their conjugates $\{\overline{\psi_n}\}$. This brings practical difficulty since evaluating conjugate potentials is hard (Korotin et al., 2019a).

Instead we parameterize potentials $\psi_n$ and $\overline{\psi_n}$ separately using input convex neural networks (ICNN) as $\psi_n^\dagger$ and $\overline{\psi_n^\ddagger}$ respectively. We add an additional cycle-consistency regularizer to enfore the conjugacy of $\psi_n^\dagger$ and $\overline{\psi_n^\ddagger}$ as in Korotin et al. (2019a). This regularizer is defined as

$$\mathcal{R}_2^{\mathbb{P}_n}(\psi_n^\dagger, \overline{\psi_n^\ddagger}) \stackrel{\text{def}}{=} \int_{\mathbb{R}^D} \|\nabla\overline{\psi_n^\ddagger} \circ \nabla\psi_n^\dagger(x) - x\|_2^2 \, d\mathbb{P}_n(x) = \|\nabla\overline{\psi_n^\ddagger} \circ \nabla\psi_n^\dagger - \text{id}_{\mathbb{R}^D}\|_{\mathbb{P}_n}^2.$$

Note that $\mathcal{R}_2^{\mathbb{P}_n}(\psi_n^\dagger, \overline{\psi_n^\ddagger}) = 0$ this condition is necessary for $\psi_n^\dagger$ and $\overline{\psi_n^\ddagger}$ to be conjugate with each other. Also, it is a sufficient condition for convex functions to be conjugates up to an additive constant.

We use one-sided regularization. In our case, computing the regularizer of the other direction $\|\nabla\psi_n^\dagger \circ \nabla\overline{\psi_n^\ddagger} - \text{id}_{\mathbb{R}^D}\|_{\overline{\mathbb{P}}}^2$ is infeasible, since $\overline{\mathbb{P}}$ is unknown. If fact, Korotin et al. (2019a) demonstrates that such one-sided condition is sufficient.

In this way we use $2N$ input convex neural networks for $\{\psi_n^\dagger, \overline{\psi_n^\ddagger}\}$. By adding the new cycle consistency regularizer into (13), we obtain our final objective:

$$\min_{\{\psi_n^\dagger, \psi_n^\ddagger\}} \sum_{n=1}^{N} \left[ \alpha_n \int_{\mathbb{R}^D} \overbrace{[\langle x, \nabla\psi_n^\dagger(x)\rangle - \underbrace{\overline{\psi_n^\ddagger}(\nabla\psi_n^\dagger(x))}_{\approx \psi_n^\ddagger(x)}]}^{\text{Approximate multiple correlation}} d\mathbb{P}_n(x) \right] + \underbrace{\tau \cdot \mathcal{R}_1^{\widehat{\mathbb{P}}}(\{\overline{\psi_n^\ddagger}\})}_{\text{Congruence reg.}} + \lambda \underbrace{\sum_{n=1}^{N} \alpha_n \mathcal{R}_2^{\mathbb{P}_n}(\psi_n^\dagger, \overline{\psi_n^\ddagger})}_{\text{Cycle regularizer}}. \quad (14)$$

Note that we express the aproximate multiple correlation by using both potentials $\{\psi_n^\dagger\}$ and $\{\overline{\psi^\ddagger}\}$. This is done to eliminate the freedom of an additive constant on $\{\psi_n^\dagger\}$ that is not addressed by cycle regularization. We denote the entire objective as $\text{MultiCorr}(\{\mathbb{P}_n\} \mid \{\psi^\dagger\}, \{\overline{\psi^\ddagger}\}; \tau, \widehat{\mathbb{P}}, \lambda)$. Analogous to Theorem 4.1, we have following result showing that this new objective enjoys the same properties as the unregularized version from (8).

**Theorem 4.2.** *Let* $\overline{\mathbb{P}} \in \mathcal{P}_{2,ac}(\mathbb{R}^D)$ *be the barycenter of* $\mathbb{P}_1, \ldots, \mathbb{P}_N \in \mathcal{P}_{2,ac}(\mathbb{R}^D)$ *w.r.t. weights* $\alpha_1, \ldots, \alpha_N$. *Let* $\{\psi_n^*\}$ *be the optimal congruent potentials of the barycenter problem. Suppose we have* $\tau, \widehat{\mathbb{P}}$ *such that* $\tau \geq 1$ *and* $\tau \cdot \widehat{\mathbb{P}} \geq \overline{\mathbb{P}}$. *Suppose we have convex potentials* $\{\psi_n^\dagger\}$ *and* $\beta^\ddagger$*-strongly convex and* $\mathcal{B}^\ddagger$*-smooth convex potentials* $\{\overline{\psi_n^\ddagger}\}$ *with* $0 < \beta^\ddagger \leq \mathcal{B}^\ddagger < \infty$ *and* $\lambda > \frac{\mathcal{B}^\dagger}{2(\beta^\ddagger)^2}$. *Then*

$$\text{MultiCorr}(\{\alpha_n, \mathbb{P}_n\} \mid \{\psi_n^\dagger\}, \{\overline{\psi_n^\ddagger}\}; \tau, \widehat{\mathbb{P}}, \lambda) \geq \text{MultiCorr}(\{\alpha_n, \mathbb{P}_n\} \mid \{\psi_n^*\}). \quad (15)$$

*Denote* $\Delta = \text{MultiCorr}(\{\alpha_n, \mathbb{P}_n\} \mid \{\psi_n^\dagger\}, \{\overline{\psi_n^\ddagger}\}; \tau, \widehat{\mathbb{P}}, \lambda) - \text{MultiCorr}(\{\alpha_n, \mathbb{P}_n\} \mid \{\psi_n^*\})$. *Then for all* $n \in \{1, \ldots, N\}$, *we have*

$$\mathbb{W}_2^2(\nabla\psi_n^\dagger \sharp \mathbb{P}_n, \overline{\mathbb{P}}) \leq \frac{2\Delta}{\alpha_n} \cdot \left( \sqrt{\frac{1}{\beta^\ddagger}} + \sqrt{\frac{1}{\lambda(\beta^\ddagger)^2 - \frac{\mathcal{B}^\dagger}{2}}} \right)^2 = O(\Delta). \quad (16)$$

Informally, Theorem 4.2 states that the better we solve the regularized dual problem, (14) the closer we expect each $\nabla\psi_n^\dagger \sharp \mathbb{P}_n$ to be to the true barycenter $\overline{\mathbb{P}}$ in $\mathbb{W}_2$. It follows from (15) that our final objective (14) is *unbiased*: the optimal solution is obtained by $\{\psi_n^*, \overline{\psi_n^*}\}$.

### 4.4 PRACTICAL ASPECTS AND OPTIMIZATION PROCEDURE

In practice, even if the choice of $\tau, \widehat{\mathbb{P}}$ does not satisfy $\tau \cdot \widehat{\mathbb{P}} \geq \overline{\mathbb{P}}$, we observe the pushforward measures $\nabla\psi_n^\dagger \sharp \mathbb{P}_n$ often converge to $\overline{\mathbb{P}}$. To partially bridge the gap between theory and practice, we *dynamically* update the measure $\widehat{\mathbb{P}}$ so that after each optimization step we set (for $\gamma \in [0, 1]$)

$$\widehat{\mathbb{P}}' := \gamma \cdot \widehat{\mathbb{P}} + (1 - \gamma) \cdot \sum_{n=1}^{N} \alpha_n \cdot \left[ \nabla\psi^\dagger \sharp \mathbb{P}_n \right],$$

i.e., the probability measure $\widehat{\mathbb{P}}'$ is a mixture of the given initial measure $\widehat{\mathbb{P}}$ and the current barycenter estimates $\{\nabla\psi^\dagger \sharp \mathbb{P}_n\}$. For the initial $\widehat{\mathbb{P}}$ one may use the barycenter of $\{\mathcal{N}(\mu_{\mathbb{P}_n}, \Sigma_{\mathbb{P}_n})\}$. It can be efficiently computed via an iterative fixed point algorithm (Álvarez-Esteban et al., 2016; Chewi et al., 2020). During the optimization, these estimates become closer to the true barycenter and can thus improve the congruence regularizer (12).

We use mini-batch stochastic gradient descent to solve (14) where the integration is done by Monte-Carlo sampling from input measures $\{\mathbb{P}_n\}$ and regularization measure $\widehat{\mathbb{P}}$, similar to Li et al. (2020). We provide the detailed optimization procedure (Algorithm 1) and discuss its computational complexity in Appendix A. In Appendix C.3, we demonstrate that the impact of the considered regularization on our model: we show that cycle consistency and the congruence condition of the potentials are well satisfied.

## 5 EXPERIMENTS

The code is written on **PyTorch** framework and is publicly available at

We compare our method [$\mathbb{CW}_2$B] with the potential-based method [$\mathbb{CRWB}$] by Li et al. (2020) (with Wasserstein-2 distance and $\mathcal{L}^2$-regularization) and with the measure-based generative method [$\mathbb{SCW}_2$B] by Fan et al. (2020). All considered methods recover $2N$ potentials $\{\psi_n^\dagger, \overline{\psi_n^\ddagger}\} \approx \{\psi_n^*, \overline{\psi_n^*}\}$ and approximate the barycenter as pushforward measures $\{\nabla \psi_n^\dagger \sharp \mathbb{P}_n\}$. Regularization in [$\mathbb{CRWB}$] allows access to the joint density of the transport plan, a feature of their method that we do not consider here. The method [$\mathbb{SCW}_2$B] additionally outputs a generated barycenter $g \sharp \mathbb{S} \approx \overline{\mathbb{P}}$ where $g$ is the generative network and $\mathbb{S}$ is the input noise distribution.

To assess the quality of the computed barycenter, we consider the **unexplained variance percentage** defined as $\mathrm{UVP}(\tilde{\mathbb{P}}) = 100 \frac{\mathbb{W}_2^2(\tilde{\mathbb{P}}, \overline{\mathbb{P}})}{1/2 \mathrm{Var}(\overline{\mathbb{P}})} \%$. When $\mathrm{UVP} \approx 0\%$, $\tilde{\mathbb{P}}$ is a good approximation of $\overline{\mathbb{P}}$. For values $\geq 100\%$, the distribution $\tilde{\mathbb{P}}$ is undesirable: a trivial baseline $\mathbb{P}^0 = \delta_{\mathbb{E}_{\overline{\mathbb{P}}}[y]}$ achieves $\mathrm{UVP}(\mathbb{P}^0) = 100\%$. Evaluating UVP in high dimensions is infeasible: empirical estimates of $\mathbb{W}_2^2$ are unreliable due to high sample complexity (Weed et al., 2019). To overcome this issue, for barycenters given by $\nabla \psi_n^\dagger \sharp \mathbb{P}_n$ we use $\mathcal{L}^2$-UVP defined by

$$\mathcal{L}^2\text{-UVP}(\nabla \psi_n^\dagger, \mathbb{P}_n) \stackrel{\text{def}}{=} 100 \frac{\|\nabla \psi_n^\dagger - \nabla \psi_n^*\|_{\mathbb{P}_n}^2}{\mathrm{Var}(\overline{\mathbb{P}})} \% \qquad \left[ \geq \mathrm{UVP}(\nabla \psi_n^\dagger \sharp \mathbb{P}_n) \right], \qquad (17)$$

where the inequality in brackets follows from (Korotin et al., 2019a, Lemma A.2). We report the weighted average of $\mathcal{L}^2$-UVP of all pushforward measures w.r.t. the weights $\alpha_n$. For barycenters given in an implicit form $g \sharp \mathbb{S}$, we compute the **Bures-Wasserstein** UVP defined by

$$\mathrm{BW}_2^2\text{-UVP}(g \sharp \mathbb{S}) \stackrel{\text{def}}{=} 100 \frac{\mathrm{BW}_2^2(g \sharp \mathbb{S}, \overline{\mathbb{P}})}{\frac{1}{2} \mathrm{Var}(\overline{\mathbb{P}})} \% \qquad \left[ \leq \mathrm{UVP}(g \sharp \mathbb{S}) \right], \qquad (18)$$

where $\mathrm{BW}_2^2(\mathbb{P}, \mathbb{Q}) = \mathbb{W}_2^2\big(\mathcal{N}(\mu_\mathbb{P}, \Sigma_\mathbb{P}), \mathcal{N}(\mu_\mathbb{Q}, \Sigma_\mathbb{Q})\big)$ is the Bures-Wasserstein metric and we use $\mu_\mathbb{P}, \Sigma_\mathbb{P}$ to denote the mean and the covariance of a distribution $\mathbb{P}$ (Chewi et al., 2020). It is known that $\mathrm{BW}_2^2$ lower-bounds $\mathbb{W}_2^2$ (Dowson & Landau, 1982), so the inequality in the brackets of (18) follows. A detailed discussion of the adopted metrics is given in Appendix C.2.

## 5.1 HIGH-DIMENSIONAL LOCATION-SCATTER EXPERIMENTS

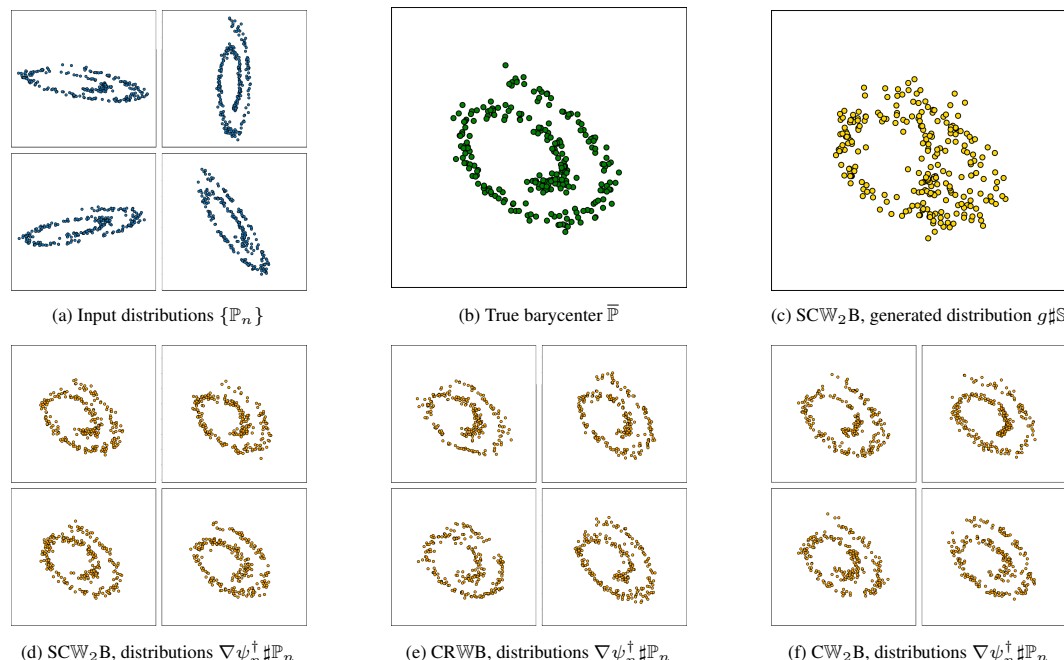

(a) Input distributions $\{\mathbb{P}_n\}$     (b) True barycenter $\overline{\mathbb{P}}$     (c) $\mathbb{SCW}_2$B, generated distribution $g \sharp \mathbb{S}$

(d) $\mathbb{SCW}_2$B, distributions $\nabla \psi_n^\dagger \sharp \mathbb{P}_n$     (e) $\mathbb{CRWB}$, distributions $\nabla \psi_n^\dagger \sharp \mathbb{P}_n$     (f) $\mathbb{CW}_2$B, distributions $\nabla \psi_n^\dagger \sharp \mathbb{P}_n$

Figure 1: Barycenter of location-scatter Swiss roll population computed by three methods.

In this section, we consider $N = 4$ with $(\alpha_1, \ldots, \alpha_4) = (0.1, 0.2, 0.3, 0.4)$ as weights. We consider the **location-scatter family** of distributions (Álvarez-Esteban et al., 2016, §4) whose true barycenter can be computed. Let $\mathbb{P}_0 \in \mathcal{P}_{2,ac}$ and define the following location-scatter family of distributions $\mathcal{F}(\mathbb{P}_0) = \{f_{S,u} \sharp \mathbb{P}_0 \mid S \in \mathcal{M}_{D \times D}^+, u \in \mathbb{R}^D\}$, where $f_{S,u} : \mathbb{R}^D \to \mathbb{R}^D$ is a linear map $f_{S,u}(x) = Sx + u$ with positive definite matrix $S \in \mathcal{M}_{D \times D}^+$. When $\{\mathbb{P}_n\} \subset \mathcal{F}(\mathbb{P}_0)$, their barycenter $\overline{\mathbb{P}}$ is also an element of $\mathcal{F}(\mathbb{P}_0)$ and can be computed via fixed-point iterations (Álvarez-Esteban et al., 2016).

Figure 1a shows a 2-dimensional location-scatter family generated by using the Swiss roll distribution as $\mathbb{P}_0$. The true barycenter is shown in Figure 1b. The generated barycenter $g \sharp \mathbb{S}$ of [SC$\mathbb{W}_2$B] is given in Figure 1c. The pushforward measures $\nabla \psi_n^\dagger \sharp \mathbb{P}_n$ of each method are provided in Figures 1d, 1e, 1f, respectively. In this example, the pushforward measures $\nabla \psi_n \sharp \mathbb{P}_n$ all reasonably approximate $\overline{\mathbb{P}}$, whereas the generated barycenter $g \sharp \mathbb{S}$ of [SC$\mathbb{W}_2$B] (Figure 1c) visibly underfits.

For quantitative comparison, we consider two choices for $\mathbb{P}_0$: the $D$-dimensional standard Gaussian distribution and the uniform distribution on $[-\sqrt{3}, +\sqrt{3}]^D$. Each $\mathbb{P}_n$ is constructed as $f_{S_n^T \Lambda S_n, 0} \sharp \mathbb{P}_0 \in \mathcal{F}(\mathbb{P}_0)$, where $S_n$ is a random rotation matrix and $\Lambda$ is diagonal with entries $[\frac{1}{2}b^0, \frac{1}{2}b^1, \ldots, 2]$ where $b = {}^{D-1}\sqrt{4}$. We consider only centered distributions (i.e. zero mean) because the barycenter of non-centered $\{\mathbb{P}_n\} \in \mathcal{P}_{2,ac}(\mathbb{R}^D)$ is the barycenter of $\{\mathbb{P}_n'\}$ shifted by $\sum_{n=1}^N \alpha_n \mu_{\mathbb{P}_n}$, where $\{\mathbb{P}_n'\}$ are centered copies of $\{\mathbb{P}_n\}$ (Álvarez-Esteban et al., 2016). Results are shown in Table 1 and 2.

In these experiments, our method outperforms [CR$\mathbb{W}$B] and [SC$\mathbb{W}_2$B]. For [CR$\mathbb{W}$B], dimension $\sim 16$ is the breakpoint: the method does not scale well to higher dimensions. [SC$\mathbb{W}_2$B] scales with the increasing dimension better, but its errors $\mathcal{L}^2$-UVP and B$\mathbb{W}_2^2$-UVP are twice as high as ours. This is likely due to the generative approximation and the difficult min-max-min optimization in [SC$\mathbb{W}_2$B]. For completeness, we also compare our algorithm to the proposed in Cuturi & Doucet (2014) which approximates the barycenter by a discrete distribution on a fixed number of free-support points. In our experiment, similar to Li et al. (2020), we set 5000 as the support size. As expected, the B$\mathbb{W}_2^2$-UVP error of the method increases drastically as the dimension grows and the method is outperformed by our approach.

To show the scalability of our method with the number of input distributions $N$, we conduct an analogous experiment with a high-dimensional location-scatter family for $N = 20$. We set $\alpha_n = \frac{2n}{N(N+1)}$ for $n = 1, 2, ..., 20$ and choose the uniform distribution on $[-\sqrt{3}, +\sqrt{3}]^D$ as $\mathbb{P}_0$ and construct distributions $\mathbb{P}_n \in \mathcal{F}(\mathbb{P}_0)$ as before. The results for dimensions 32, 64 and 128 are provided in Table 3. Similar to the results from Tables 1 and 2, we see that our method outperforms the alternatives.

| Metric | Method | D=2 | 4 | 8 | 16 | 32 | 64 | 128 | 256 |
|---|---|---|---|---|---|---|---|---|---|
| B$\mathbb{W}_2^2$-UVP, % | [FC$\mathbb{W}$B], Cuturi & Doucet (2014) | 0.7 | 0.68 | 1.41 | 3.87 | 8.85 | 14.08 | 18.11 | 21.33 |
| | [SC$\mathbb{W}_2$B], (Fan et al., 2020) | 0.07 | 0.09 | 0.16 | 0.28 | 0.43 | 0.59 | 1.28 | 2.85 |
| $\mathcal{L}_2$-UVP, % | | 0.08 | 0.10 | 0.17 | 0.29 | 0.47 | 0.63 | 1.14 | 1.50 |
| (potentials) | [CR$\mathbb{W}$B], (Li et al., 2020) | 0.99 | 2.52 | 8.62 | 22.23 | 67.01 | | >100 | |
| | [C$\mathbb{W}_2$B], **ours** | 0.06 | 0.05 | 0.07 | 0.11 | 0.19 | 0.24 | 0.42 | 0.83 |

Table 1: Comparison of UVP for the case $\{\mathbb{P}_n\} \subset \mathcal{F}(\mathbb{P}_0)$, $\mathbb{P}_0 = \mathcal{N}(0, I_D)$, $N = 4$.

| Metric | Method | D=2 | 4 | 8 | 16 | 32 | 64 | 128 | 256 |
|---|---|---|---|---|---|---|---|---|---|
| B$\mathbb{W}_2^2$-UVP, % | [FC$\mathbb{W}$B], Cuturi & Doucet (2014) | 0.64 | 0.77 | 1.22 | 3.75 | 8.92 | 14.3 | 18.46 | 21.64 |
| | [SC$\mathbb{W}_2$B], (Fan et al., 2020) | 0.12 | 0.10 | 0.19 | 0.29 | 0.46 | 0.6 | 1.38 | 2.9 |
| $\mathcal{L}_2$-UVP, % | | 0.17 | 0.12 | 0.2 | 0.31 | 0.47 | 0.62 | 1.21 | 1.52 |
| (potentials) | [CR$\mathbb{W}$B], (Li et al., 2020) | 0.58 | 1.83 | 8.09 | 21.23 | 55.17 | | > 100 | |
| | [C$\mathbb{W}_2$B], **ours** | 0.17 | 0.08 | 0.06 | 0.1 | 0.2 | 0.25 | 0.42 | 0.82 |

Table 2: Comparison of UVP for the case $\{\mathbb{P}_n\} \subset \mathcal{F}(\mathbb{P}_0)$, $\mathbb{P}_0 = \text{Uniform}([-\sqrt{3}, +\sqrt{3}]^D)$, $N = 4$.

| Metric | Method | D=32 | 64 | 128 |
|---|---|---|---|---|
| B$\mathbb{W}_2^2$-UVP, % | [FC$\mathbb{W}$B], Cuturi & Doucet (2014) | 14.09 | 26.21 | 38.43 |
| | [SC$\mathbb{W}_2$B], (Fan et al., 2020) | 0.62 | 0.93 | 1.83 |
| $\mathcal{L}_2$-UVP, % | | 0.60 | 0.86 | 1.52 |
| (potentials) | [C$\mathbb{W}_2$B], **ours** | 0.31 | 0.58 | 1.45 |

Table 3: Comparison of UVP for the case $\{\mathbb{P}_n\} \subset \mathcal{F}(\mathbb{P}_0)$, $\mathbb{P}_0 = \text{Uniform}([-\sqrt{3}, +\sqrt{3}]^D)$, $N = 20$.

## 5.2 SUBSET POSTERIOR AGGREGATION

We apply our method to aggregate subset posterior distributions. The barycenter of subset posteriors converges to the true posterior (Srivastava et al., 2018). Thus, computing the barycenter of subset

posteriors is an efficient alternative to obtaining a full posterior in the big data setting (Srivastava et al., 2015; Staib et al., 2017; Li et al., 2020).

Analogous to (Li et al., 2020), we consider Poisson and negative binomial regressions for predicting the hourly number of bike rentals using features such as the day of the week and weather conditions.[2] We consider the posterior on the 8-dimensional regression coefficients for both Poisson and negative binomial regressions. We randomly split the data into $N = 5$ equally-sized subsets and obtain $10^5$ samples from each subset posterior using the Stan library (Carpenter et al., 2017). This gives the discrete uniform distributions $\{\mathbb{P}_n\}$ supported on the samples. As the ground truth barycenter $\overline{\mathbb{P}}$, we consider the full dataset posterior also consisting of $10^5$ points.

We use $\text{B}\mathbb{W}_2^2\text{-UVP}(\tilde{\mathbb{P}}, \overline{\mathbb{P}})$ to compare the estimated barycenter $\tilde{\mathbb{P}}$ (pushforward measure $\nabla\psi_n^\dagger\sharp\mathbb{P}_n$ or generated measure $g\sharp\mathbb{S}$) with the true barycenter. The results are in Table 4. All considered methods perform well (UVP< 2%), but our method outperforms the alternatives.

| | Regression | SC$\mathbb{W}_2$B, (Fan et al., 2020) | [CR$\mathbb{W}$B], (Li et al., 2020) | C$\mathbb{W}_2$B, **ours** |
|---|---|---|---|---|
| | | $\mathbb{P} = g\sharp\mathbb{S}$ | $\mathbb{P} = \nabla\psi_n\sharp\mathbb{P}_n$ | |
| B$\mathbb{W}_2^2$-UVP, % | Poisson | 0.67 | 0.41 | 1.53 | 0.1 |
| | negative binomial | 0.15 | 0.15 | 1.26 | 0.11 |

Table 4: Comparison of UVP for recovered barycenters in our subset posterior aggregation task.

### 5.3 COLOR PALETTE AVERAGING

For qualitative study, we apply our method to aggregating color palettes of images. For an RGB image $\mathcal{I}$, its color palette is defined by the discrete uniform distribution $\mathbb{P}(\mathcal{I})$ of all its pixels $\in [0,1]^3$. For 3 images $\{\mathcal{I}_n\}$ we compute the barycenter $\overline{\mathbb{P}}$ of each color palette $\mathbb{P}_n = \mathbb{P}(\mathcal{I}_n)$ w.r.t. uniform weights $\alpha_n = \frac{1}{3}$. We apply each computed potential $\nabla\psi_n^\dagger$ pixel-wise to $\mathcal{I}_n$ to obtain the "pushforward" image $\nabla\psi_n^\dagger\sharp\mathcal{I}_n$. These "pushforward" images should be close to the barycenter $\overline{\mathbb{P}}$ of $\{\mathbb{P}_n\}$.

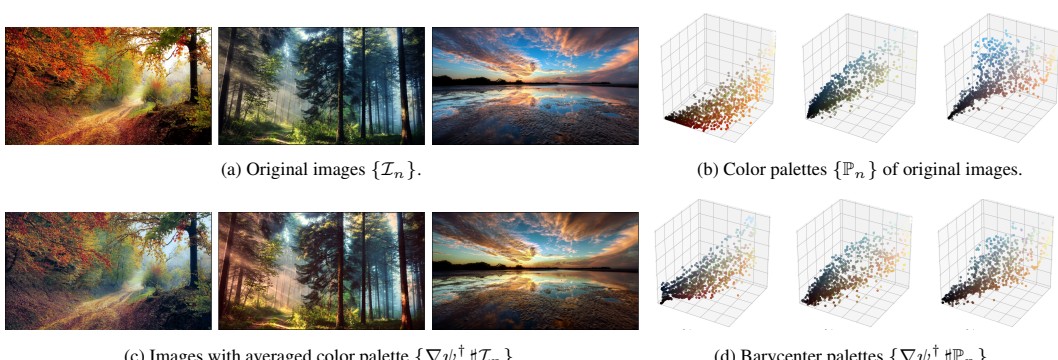

(a) Original images $\{\mathcal{I}_n\}$.   (b) Color palettes $\{\mathbb{P}_n\}$ of original images.

(c) Images with averaged color palette $\{\nabla\psi_n^\dagger\sharp\mathcal{I}_n\}$.   (d) Barycenter palettes $\{\nabla\psi_n^\dagger\sharp\mathbb{P}_n\}$.

Figure 2: Results of our method applied to averaging color palettes of images.

The results are provided in Figure 2. Note that the image $\nabla\psi_1^\dagger\sharp\mathcal{I}_1$ inherits certain attributes of images $\mathcal{I}_2$ and $\mathcal{I}_3$: the sky becomes bluer and the trees becomes greener. On the other hand, the sunlight in images $\nabla\psi_2^\dagger\sharp\mathcal{I}_2, \nabla\psi_3^\dagger\sharp\mathcal{I}_3$ has acquired an orange tint, thanks to the dominance of orange in $\mathcal{I}_1$.

### ACKNOWLEDGMENTS

The Skoltech Advanced Data Analytics in Science and Engineering Group acknowledges the support of Russian Foundation for Basic Research grant 20-01-00203, Skoltech-MIT NGP initiative and thanks the Skoltech CDISE HPC Zhores cluster staff for computing cluster provision.

The MIT Geometric Data Processing group acknowledges the generous support of Army Research Office grant W911NF2010168, of Air Force Office of Scientific Research award FA9550-19-1-031, of National Science Foundation grant IIS-1838071, from the CSAIL Systems that Learn program, from the MIT–IBM Watson AI Laboratory, from the Toyota–CSAIL Joint Research Center, from a gift from Adobe Systems, from an MIT.nano Immersion Lab/NCSOFT Gaming Program seed grant, and from the Skoltech–MIT Next Generation Program.

---

[2]http://archive.ics.uci.edu/ml/datasets/Bike+Sharing+Dataset

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

## A  THE ALGORITHM

The numerical procedure for solving our final objective (14) is given below.

---

**Algorithm 1:** Numerical Procedure for Optimizing Multiple Correlations (14)

---

**Input :** Distributions $\mathbb{P}_1, \ldots, \mathbb{P}_N$ with sample access;

Weights $\alpha_1, \ldots, \alpha_N \geq 0$ with $\sum_{n=1}^N \alpha_n = 1$;

Regularization distribution $\widehat{\mathbb{P}}'$ given by a sampler;

Congruence regularizer coefficient $\tau \geq 1$;

Balancing coefficient $\gamma \in [0, 1]$;

Cycle-consistency regularizer coefficient $\lambda > 0$;

$2N$ ICNNs $\{\psi_{\theta_n}, \overline{\psi_{\omega_n}}\}$;

Batch size $K > 0$;

**for** $t = 1, 2, \ldots$ **do**

1. Sample batches $X_n \sim \mathbb{P}_n$ for all $n = 1, \ldots, N$;

2. Compute the pushforwards $Y_n = \nabla \psi_{\theta_n} \sharp X_n$ for all $n = 1, \ldots, N$;

3. Sample batch $Y_0 \sim \widehat{\mathbb{P}}$;

4. Compute the Monte-Carlo estimate of the congruence regularizer:

$$\mathcal{L}_{\text{Congruence}} := \frac{1}{K} \cdot \sum_{n=1}^N \gamma_n \sum_{y \in Y_n} \Big[ \sum_{n'=1}^N \alpha_{n'} \overline{\psi_{\omega_{n'}}}(y) - \frac{\|y\|^2}{2} \Big]_+,$$

where $\gamma_0 = \gamma$ and $\gamma_n = \alpha_n \cdot (1 - \gamma)$ for $n = 1, 2, \ldots, N$;

5. Compute the Monte-Carlo estimate of the cycle-consistency regularizer:

$$\mathcal{L}_{\text{Cycle}} := \frac{1}{K} \sum_{n=1}^N \alpha_n \bigg[ \sum_{x \in X_n} \|\nabla \overline{\psi_{\omega_n}} \circ \nabla \psi_{\theta_n}(x) - x\|_2^2 \bigg];$$

6. Compute the Monte-Carlo estimate of multiple correlations:

$$\mathcal{L}_{\text{MultiCorr}} := \sum_{n=1}^N \bigg[ \alpha_n \cdot \frac{1}{K} \sum_{x \in X_n} \big[ \langle x, \nabla \psi_{\theta_n}(x) \rangle - \overline{\psi_{\omega_n}}(\nabla \psi_{\theta_n}(x)) \big] \bigg];$$

7. Compute the total loss:

$$\mathcal{L}_{\text{Total}} := \mathcal{L}_{\text{MultiCorr}} + \lambda \cdot \mathcal{L}_{\text{Cycle}} + \tau \cdot \mathcal{L}_{\text{Congruence}};$$

8. Perform a gradient step over $\{\theta_n, \omega_n\}$ by using $\frac{\partial \mathcal{L}_{\text{Total}}}{\partial \{\theta_n, \omega_n\}}$;

**end**

---

**Parametrization of the potentials.** To parametrize potentials $\{\psi_{\theta_n}, \overline{\psi_{\omega_n}}\}$, we use DenseICNN (dense input convex neural network) with quadratic skip connections; see (Korotin et al., 2019a, Appendix B.2). As an initialization step, we pre-train the potentials to satisfy

$$\psi_{\theta_n}(x) \approx \frac{\|x\|^2}{2} \qquad \text{and} \qquad \overline{\psi_{\omega_n}}(y) \approx \frac{\|y\|^2}{2}.$$

Such pre-training provides a good start for the networks: each $\psi_{\theta_n}$ is approximately conjugate to the corresponding $\overline{\psi_{\omega_n}}$. On the other hand, the initial networks $\{\psi_{\theta_n}\}$ are approximate congruent according to (5).

**Computational Complexity.** For a single training iteration, the time complexity of both forward (evaluation) and backward (computing the gradient with respect to the parameters) passes through the objective function (14) is $O(NT)$. Here $N$ is the number of input distributions and $T$ is the time taken by evaluating each individual potential (parameterized as a neural network) on a batch of points sampled from either $\mathbb{P}_n$ or $\widehat{\mathbb{P}}$. This claim follows from the well-known fact that gradient evaluation $\nabla_\theta h_\theta(x)$ of $h_\theta : \mathbb{R}^D \to \mathbb{R}$, when parameterized as a neural network, requires time proportional

to the size of the computational graph. Hence, gradient computation requires computational time proportional to the time for evaluating the function $h_\theta(x)$ itself. The same holds when computing the derivative with respect to $x$. Then, for instance, computing the term $\nabla\overline{\psi_n^{\ddagger}} \circ \nabla\psi_n^{\dagger}(x)$ in (14) takes $O(T)$ time. The gradient of this term with respect to $\theta$ also takes $O(T)$ time: Hessian-vector products that appear can be calculated in $O(T)$ time using the famous Hessian trick, see Pearlmutter (1994).

In practice, we compute all the gradients using automatic differentiation. We empirically measured that for our DenseICNN potentials, the computation of their gradient w.r.t. input $x$, i.e., $\nabla\psi^{\dagger}(x)$, requires roughly 3-4x more time than the computation of $\psi^{\dagger}(x)$.

## B  PROOFS

In this section, we prove our main Theorems 4.1 and 4.2.

We use $\mathcal{L}^2(\mathbb{R}^D \to \mathbb{R}^D, \mu)$ to denote the **Hilbert space** of functions $f : \mathbb{R}^D \to \mathbb{R}^D$ with integrable square w.r.t. a probability measure $\mu$. The corresponding inner product for $f_1, f_2 \in \mathcal{L}^2(\mathbb{R}^D \to \mathbb{R}^D, \mu)$ is denoted by

$$\langle f_1, f_2 \rangle_\mu \stackrel{\text{def}}{=} \int_{\mathbb{R}^D} \langle f_1(x), f_2(x) \rangle d\mu(x),$$

where $\langle f_1(x), f_2(x) \rangle$ is the Euclidean dot product. We use $\| \cdot \|_\mu = \sqrt{\langle \cdot, \cdot \rangle_\mu}$ to denote the norm induced by the inner product in $\mathcal{L}^2(\mathbb{R}^D \to \mathbb{R}^D, \mu)$.

We also recall a useful property of lower semi-continuous convex function $\psi : \mathbb{R}^D \to \mathbb{R}$:

$$\nabla\psi(x) = \arg\max_{y \in \mathbb{R}^D} \left[ \langle y, x \rangle - \overline{\psi}(y) \right], \tag{19}$$

which follows from the fact that

$$\hat{y} = \arg\max_{y \in \mathbb{R}^D} \left[ \langle y, x \rangle - \overline{\psi}(y) \right] \iff x - \nabla\overline{\psi}(\hat{y}) = 0.$$

We begin with the proof of Theorem 4.1.

*Proof.* We consider the difference between the estimated correlations and true ones:

$$\Delta = \sum_{n=1}^{N} \alpha_n \int_{\mathbb{R}^D} \psi_n(x) d\mathbb{P}_n(x) - \sum_{n=1}^{N} \alpha_n \int_{\mathbb{R}^D} \psi_n^*(x) d\mathbb{P}_n(x) =$$

$$\sum_{n=1}^{N} \alpha_n \int_{\mathbb{R}^D} \left[ \langle \nabla\psi_n(x), x \rangle - \overline{\psi_n}(\nabla\psi_n(x)) \right] d\mathbb{P}_n(x) -$$

$$\sum_{n=1}^{N} \alpha_n \int_{\mathbb{R}^D} \left[ \langle \nabla\psi_n^*(x), x \rangle - \overline{\psi_n^*}(\nabla\psi_n^*(x)) \right] d\mathbb{P}_n(x), \tag{20}$$

where we twice use (19) for $f = \psi_n$ and $f = \psi_n^*$. We note that

$$\sum_{n=1}^{N} \alpha_n \int_{\mathbb{R}^D} \langle \nabla\psi_n^*(x), x \rangle d\mathbb{P}_n(x) = \sum_{n=1}^{N} \alpha_n \int_{\mathbb{R}^D} \langle y, \nabla\overline{\psi_n^*}(y) \rangle d\overline{\mathbb{P}}(y) =$$

$$\int_{\mathbb{R}^D} \langle y, \sum_{n=1}^{N} \alpha_n \nabla\overline{\psi_n^*}(y) \rangle d\overline{\mathbb{P}}(y) = \int_{\mathbb{R}^D} \langle y, y \rangle d\overline{\mathbb{P}}(y) = \|\text{id}_{\mathbb{R}^D}\|_{\overline{\mathbb{P}}}^2, \tag{21}$$

where we use of change-of-variable formula for $\nabla\psi_n^* \sharp \mathbb{P}_n = \overline{\mathbb{P}}$ and (5). Analogously,

$$\sum_{n=1}^{N} \alpha_n \int_{\mathbb{R}^D} \overline{\psi_n^*}(\nabla\psi_n^*(x)) d\mathbb{P}_n(x) = \sum_{n=1}^{N} \alpha_n \int_{\mathbb{R}^D} \overline{\psi_n^*}(y) d\overline{\mathbb{P}}(y) =$$

$$\int_{\mathbb{R}^D} \sum_{n=1}^{N} \alpha_n \overline{\psi_n^*}(y) d\overline{\mathbb{P}}(y) = \int_{\mathbb{R}^D} \frac{\|y\|^2}{2} d\overline{\mathbb{P}}(y) = \frac{1}{2} \|\text{id}_{\mathbb{R}^D}\|_{\overline{\mathbb{P}}}^2. \tag{22}$$

Since each $\psi_n$ is $\mathcal{B}$-smooth, we conclude that $\overline{\psi_n}$ is $\frac{1}{\mathcal{B}}$-strongly convex, see (Kakade et al., 2009). Thus, we have

$$\overline{\psi_n}\big(\nabla\psi_n^*(x))) \geq$$

$$\overline{\psi_n}\big(\nabla\psi_n(x))) + \langle\underbrace{\nabla\overline{\psi_n^\dagger}\circ\nabla\psi_n^\dagger(x)}_{=x}, \nabla\psi_n^*(x) - \nabla\psi_n(x)\rangle + \frac{1}{2\mathcal{B}}\|\nabla\psi_n^*(x) - \nabla\psi_n(x)\|^2 =$$

$$\overline{\psi_n}\big(\nabla\psi_n(x))) + \langle x, \nabla\psi_n^*(x) - \nabla\psi_n(x)\rangle + \frac{1}{2\mathcal{B}}\|\nabla\psi_n^*(x) - \nabla\psi_n(x)\|^2, \quad (23)$$

or equivalently

$$-\overline{\psi_n}\big(\nabla\psi_n(x))) \geq -\overline{\psi_n}\big(\nabla\psi_n^*(x))) + \langle x, \nabla\psi_n^*(x) - \nabla\psi_n(x)\rangle + \frac{1}{2\mathcal{B}}\|\nabla\psi_n^*(x) - \nabla\psi_n(x)\|^2. \quad (24)$$

We integrate (24) w.r.t. $\mathbb{P}_n$ and sum over $n = 1, 2, \ldots, N$ with weights $\alpha_n$:

$$-\sum_{n=1}^{N}\alpha_n\int_{\mathbb{R}^D}\overline{\psi_n}\big(\nabla\psi_n(x))d\mathbb{P}_n(x) \geq$$

$$-\sum_{n=1}^{N}\alpha_n\int_{\mathbb{R}^D}\overline{\psi_n}(\nabla\psi_n^*(x))d\mathbb{P}_n(x) + \sum_{n=1}^{N}\alpha_n\langle x, \nabla\psi_n^*(x)\rangle_{\mathbb{P}_n} - \sum_{n=1}^{N}\alpha_n\langle x, \nabla\psi_n(x)\rangle_{\mathbb{P}_n} +$$

$$\sum_{n=1}^{N}\alpha_n\frac{1}{2\mathcal{B}}\|\nabla\psi_n^*(x) - \nabla\psi_n(x)\|_{\mathbb{P}_n}^2 =$$

$$-\int_{\mathbb{R}^D}\sum_{n=1}^{N}\alpha_n\overline{\psi_n}\big(y)d\overline{\mathbb{P}}(y) + \sum_{n=1}^{N}\alpha_n\langle x, \nabla\psi_n^*(x)\rangle_{\mathbb{P}_n} -$$

$$\sum_{n=1}^{N}\alpha_n\langle x, \nabla\psi_n(x)\rangle_{\mathbb{P}_n} + \sum_{n=1}^{N}\alpha_n\frac{1}{2\mathcal{B}}\|\nabla\psi_n^*(x) - \nabla\psi_n(x)\|_{\mathbb{P}_n}^2. \quad (25)$$

We note that

$$-\int_{\mathbb{R}^D}\sum_{n=1}^{N}\alpha_n\overline{\psi_n}\big(y)d\overline{\mathbb{P}}(y) = \int_{\mathbb{R}^D}\big[\frac{\|y\|^2}{2} - \sum_{n=1}^{N}\alpha_n\overline{\psi_n}\big(y)\big]d\overline{\mathbb{P}}(y) - \int_{\mathbb{R}^D}\frac{\|y\|^2}{2}d\overline{\mathbb{P}}(y)$$

$$\int_{\mathbb{R}^D}\big[\frac{\|y\|^2}{2} - \sum_{n=1}^{N}\alpha_n\overline{\psi_n}\big(y)\big]d\overline{\mathbb{P}}(y) - \frac{1}{2}\|\mathrm{id}_{\mathbb{R}^D}\|_{\overline{\mathbb{P}}}^2. \quad (26)$$

Now we substitute (25), (26), (21) and (22) into (20) to obtain (9). $\qquad\square$

Next, we prove Theorem 4.2.

*Proof.* Since $\overline{\psi_n^\ddagger}$ is $\beta^\ddagger$ strongly convex, its conjugate $\psi_n^\ddagger$ is $\frac{1}{\beta^\ddagger}$-smooth, i.e. has $\frac{1}{\beta^\ddagger}$-Lipschitz gradient $\nabla\psi_n^\ddagger$ (Kakade et al., 2009). Thus, for all $x, x' \in \mathbb{R}^D$:

$$\|\nabla\psi_n^\ddagger(x) - \nabla\psi_n^\ddagger(x')\|^2 \leq (\frac{1}{\beta^\ddagger})^2 \cdot \|x - x'\|^2.$$

We substitute $x' = \nabla\overline{\psi_n^\ddagger} \circ \nabla\psi_n^\dagger(y) = \big(\nabla\psi_n^\ddagger\big)^{-1} \circ \nabla\psi_n^\dagger(y)$ and obtain:

$$\|\nabla\psi_n^\dagger(x) - \nabla\psi_n^\ddagger(x)\|^2 \leq (\frac{1}{\beta^\ddagger})^2\|x - \nabla\overline{\psi_n^\ddagger} \circ \nabla\psi_n^\dagger(x)\|^2. \quad (27)$$

Since the function $\overline{\psi_n^\ddagger}$ is $\mathcal{B}^\ddagger$-smooth, we have for all $x \in \mathbb{R}^D$:

$$\overline{\psi_n^\ddagger}(\nabla\psi_n^\dagger(x)) \leq \overline{\psi_n^\ddagger}(\nabla\psi_n^\ddagger(x)) + \langle\underbrace{\nabla\overline{\psi_n^\ddagger} \circ \nabla\psi_n^\ddagger(x)}_{=x}, \nabla\psi_n^\dagger(x) - \nabla\psi_n^\ddagger(x)\rangle + \frac{\mathcal{B}^\ddagger}{2}\|\nabla\psi_n^\dagger(x) - \nabla\psi_n^\ddagger(x)\|^2,$$

that is equivalent to:

$$\langle x, \nabla\psi_n^\dagger(x)\rangle - \overline{\psi_n^\ddagger}(\nabla\psi_n^\dagger(x)) \geq \underbrace{\langle x, \nabla\psi_n^\ddagger(x)\rangle - \overline{\psi_n^\ddagger}(\nabla\psi_n^\ddagger(x))}_{\psi_n^\ddagger(x)} - \frac{\mathcal{B}^\ddagger}{2}\|\nabla\psi_n^\dagger(x) - \nabla\psi_n^\ddagger(x)\|^2. \quad (28)$$

We combine (28) with (27) to obtain

$$\langle x, \nabla\psi_n^\dagger(x)\rangle - \overline{\psi_n^\ddagger}(\nabla\psi_n^\dagger(x)) \geq \psi_n^\ddagger(x) - \frac{\mathcal{B}^\ddagger}{2(\beta^\ddagger)^2}\cdot\|\mathrm{id}_{\mathbb{R}^D} - \nabla\overline{\psi_n^\ddagger}\circ\nabla\psi_n^\dagger\|^2. \quad (29)$$

For every $n = 1, 2, \ldots, N$ we integrate (29) w.r.t. $\mathbb{P}_n$ and sum up the corresponding cycle-consistency regularization term:

$$\int_{\mathbb{R}^D}\big[\langle x, \nabla\psi_n^\dagger(x)\rangle - \overline{\psi_n^\ddagger}(\nabla\psi_n^\dagger(x))\big]d\mathbb{P}_n(x) + \lambda\cdot\|\nabla\overline{\psi_n^\ddagger}\circ\nabla\psi_n^\dagger - \mathrm{id}_{\mathbb{R}^D}\|_{\mathbb{P}_n}^2 \geq$$
$$\int_{\mathbb{R}^D}\psi^\ddagger(x)d\mathbb{P}_n(x) + \big(\lambda - \frac{\mathcal{B}^\dagger}{2(\beta^\ddagger)^2}\big)\cdot\underbrace{\|\nabla\overline{\psi_n^\ddagger}\circ\nabla\psi_n^\dagger - \mathrm{id}_{\mathbb{R}^D}\|_{\mathbb{P}_n}^2}_{\mathcal{R}_2^{\mathbb{P}_n}(\psi_n^\dagger, \overline{\psi_n^\ddagger})}. \quad (30)$$

We sum (30) for $n = 1, 2, \ldots, N$ w.r.t. weights $\alpha_n$ to obtain:

$$\sum_{n=1}^N \alpha_n \int_{\mathbb{R}^D}\big[\langle x, \nabla\psi_n^\dagger(x)\rangle - \overline{\psi_n^\ddagger}(\nabla\psi_n^\dagger(x))\big]d\mathbb{P}_n(x) + \lambda\sum_{n=1}^N\alpha_n\mathcal{R}_2^{\mathbb{P}_n}(\psi_n^\dagger, \overline{\psi_n^\ddagger}) \geq$$
$$\underbrace{\sum_{n=1}^N\alpha_n\int_{\mathbb{R}^D}\psi^\ddagger(x)d\mathbb{P}_n(x)}_{\mathrm{MultiCorr}(\{\alpha_n, \mathbb{P}_n\}|\{\psi_n^\ddagger\})} + \sum_{n=1}^N\alpha_n\big(\lambda - \frac{\mathcal{B}^\dagger}{2(\beta^\ddagger)^2}\big)\cdot\mathcal{R}_2^{\mathbb{P}_n}(\psi_n^\dagger, \overline{\psi_n^\ddagger}).$$

We add $\tau\cdot\mathcal{R}_1^{\widehat{\mathbb{P}}}(\{\overline{\psi_n^\ddagger}\})$ to both sides of (31) to get

$$\mathrm{MultiCorr}\big(\{\alpha_n, \mathbb{P}_n\}\mid\{\psi_n^\dagger\}, \{\overline{\psi_n^\ddagger}\}; \tau, \widehat{\mathbb{P}}, \lambda\big) \geq \mathrm{MultiCorr}(\{\alpha_n, \mathbb{P}_n\}\mid\{\psi_n^\ddagger\}) +$$
$$\tau\cdot\mathcal{R}_1^{\widehat{\mathbb{P}}}(\{\overline{\psi_n^\ddagger}\}) + \sum_{n=1}^N\alpha_n\big(\lambda - \frac{\mathcal{B}^\dagger}{2(\beta^\ddagger)^2}\big)\cdot\mathcal{R}_2^{\mathbb{P}_n}(\psi_n^\dagger, \overline{\psi_n^\ddagger}). \quad (31)$$

We substract $\mathrm{MultiCorr}(\{\alpha_n, \mathbb{P}_n\}\mid\{\psi_n^\ddagger\})$ from both sides and use Theorem 4.1 to obtain

$$\Delta \geq -\int_{\mathbb{R}^D}\sum_{n=1}^N\big[\alpha_n\overline{\psi_n^\ddagger}(y) - \frac{\|y\|^2}{2}\big]d\overline{\mathbb{P}}(y) + \frac{\beta^\ddagger}{2}\sum_{n=1}^N\alpha_n\|\nabla\psi_n^*(x) - \nabla\psi_n^\ddagger(x)\|_{\mathbb{P}_n}^2 + \quad (32)$$

$$\tau\cdot\mathcal{R}_1^{\widehat{\mathbb{P}}}(\{\overline{\psi_n^\ddagger}\}) + \sum_{n=1}^N\alpha_n\big(\lambda - \frac{\mathcal{B}^\dagger}{2(\beta^\ddagger)^2}\big)\cdot\mathcal{R}_2^{\mathbb{P}_n}(\psi_n^\dagger, \overline{\psi_n^\ddagger}) \geq \quad (33)$$

$$\sum_{n=1}^N\alpha_n\big(\lambda - \frac{\mathcal{B}^\dagger}{2(\beta^\ddagger)^2}\big)\cdot\mathcal{R}_2^{\mathbb{P}_n}(\psi_n^\dagger, \overline{\psi_n^\ddagger}) + \frac{\beta^\ddagger}{2}\sum_{n=1}^N\alpha_n\|\nabla\psi_n^*(x) - \nabla\psi_n^\ddagger(x)\|_{\mathbb{P}_n}^2. \quad (34)$$

In transition from (33) to (34), we explot the fact that the sum of the first term of (32) with the regularizer $\tau\cdot\mathcal{R}_1^{\widehat{\mathbb{P}}}(\{\overline{\psi_n^\ddagger}\})$. Since $\lambda > \frac{\mathcal{B}^\dagger}{2(\beta^\ddagger)^2}$, from (34) we immediately conclude $\Delta \geq 0$; i.e., the **multiple correlations upper bound** (15) holds true. On the other hand, for every $n = 1, 2, \ldots, N$ we have

$$\|\nabla\psi_n^*(x) - \nabla\psi_n^\ddagger(x)\|_{\mathbb{P}_n}^2 \leq \frac{2\Delta}{\alpha_n\beta^\ddagger} \quad \text{and} \quad \|\nabla\overline{\psi_n^\ddagger}\circ\nabla\psi_n^\dagger - \mathrm{id}_{\mathbb{R}^D}\|_{\mathbb{P}_n}^2 \leq \frac{2\Delta}{\alpha_n\cdot(\lambda - \frac{\mathcal{B}^\dagger}{2(\beta^\ddagger)^2})}. \quad (35)$$

We combine the second part of (35) with (27) integrated w.r.t. $\mathbb{P}_n$:

$$\|\nabla\psi_n^\ddagger - \nabla\psi_n^\dagger\|_{\mathbb{P}_n}^2 \leq \frac{2\Delta}{\alpha_n\cdot(\lambda(\beta^\ddagger)^2 - \frac{\mathcal{B}^\dagger}{2})}. \quad (36)$$

Finally, we use the triangle inequality for $\|\cdot\|_{\mathbb{P}_n}$ and conclude

$$\|\nabla\psi_n^* - \nabla\psi_n^\dagger\|_{\mathbb{P}_n} \leq \|\nabla\psi_n^\ddagger - \nabla\psi_n^\dagger\|_{\mathbb{P}_n} + \|\nabla\psi_n^\ddagger - \nabla\psi_n^*\|_{\mathbb{P}_n} \leq$$

$$\sqrt{\frac{2\Delta}{\alpha_n}} \cdot \left(\sqrt{\frac{1}{\beta^\ddagger}} + \sqrt{\frac{1}{\lambda(\beta^\ddagger)^2 - \frac{\mathcal{B}^\dagger}{2}}}\right), \tag{37}$$

i.e.,

$$\mathbb{W}_2^2(\nabla\psi_n^\dagger\sharp\mathbb{P}_n, \overline{\mathbb{P}}) \leq \|\nabla\psi_n^* - \nabla\psi_n^\dagger\|_{\mathbb{P}_n}^2 \leq \frac{2\Delta}{\alpha_n} \cdot \left(\sqrt{\frac{1}{\beta^\ddagger}} + \sqrt{\frac{1}{\lambda(\beta^\ddagger)^2 - \frac{\mathcal{B}^\dagger}{2}}}\right)^2 = O(\Delta),$$

where the first inequality follows from (Korotin et al., 2019a, Lemma A.2). $\qquad\square$

## C  EXPERIMENTAL DETAILS AND EXTRA RESULTS

In this section, we provide experimental details and additional results. In Subsection C.1, we demonstrate qualitative results of computed barycenters in the 2-dimensional space. In Subsection C.2, we discuss used metrics in more detail. In Subsection C.4, we list the used hyperparameters of our method ($\mathbb{CW}_2\text{B}$) and methods [$\text{SCW}_2\text{B}$], [$\text{CRWB}$].

### C.1  ADDITIONAL TOY EXPERIMENTS IN 2D

We provide additional qualitative examples of computed barycenters of probability measures on $\mathbb{R}^2$.

In Figure 3, we consider the location-scatter family $\mathcal{F}(\mathbb{P}_0)$ with $\mathbb{P}_0 = \text{Uniform}[-\sqrt{3}, \sqrt{3}]^D$. In principle, all the methods capture the true barycenter. However, the generated distribution $g\sharp\mathbb{S}$ of [$\text{SCW}_2\text{B}$] (Figure 3c) provides samples that lies outside of the actual barycenter's support (Figure 3b). Also, in [$\text{CRWB}$] method, one of the potentials' pushforward measure (top-right in Figure 3e) has visual artifacts.

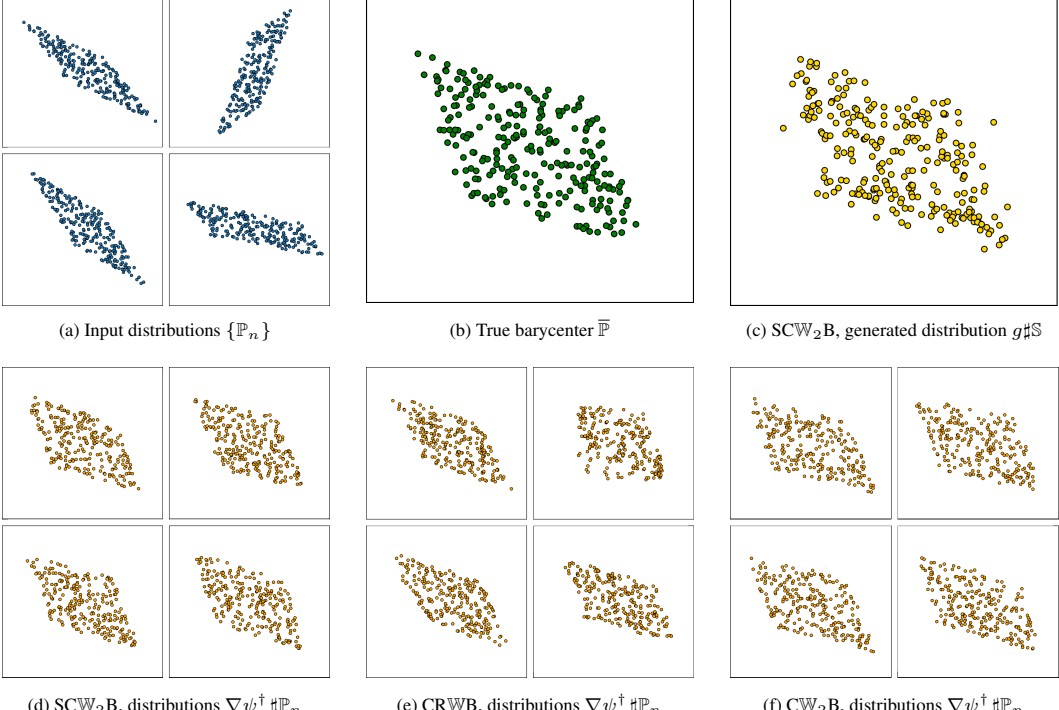

(a) Input distributions $\{\mathbb{P}_n\}$       (b) True barycenter $\overline{\mathbb{P}}$       (c) $\text{SCW}_2\text{B}$, generated distribution $g\sharp\mathbb{S}$

(d) $\text{SCW}_2\text{B}$, distributions $\nabla\psi_n^\dagger\sharp\mathbb{P}_n$       (e) $\text{CRWB}$, distributions $\nabla\psi_n^\dagger\sharp\mathbb{P}_n$       (f) $\text{CW}_2\text{B}$, distributions $\nabla\psi_n^\dagger\sharp\mathbb{P}_n$

Figure 3: Barycenter of a random location-scatter population computed by different methods.

In Figure 4, we consider the Gaussian Mixture example by (Fan et al., 2020). The barycenter computed by [$\text{SCW}_2\text{B}$] method (Figure 4b) suffers from the behavior similar to mode collapse.

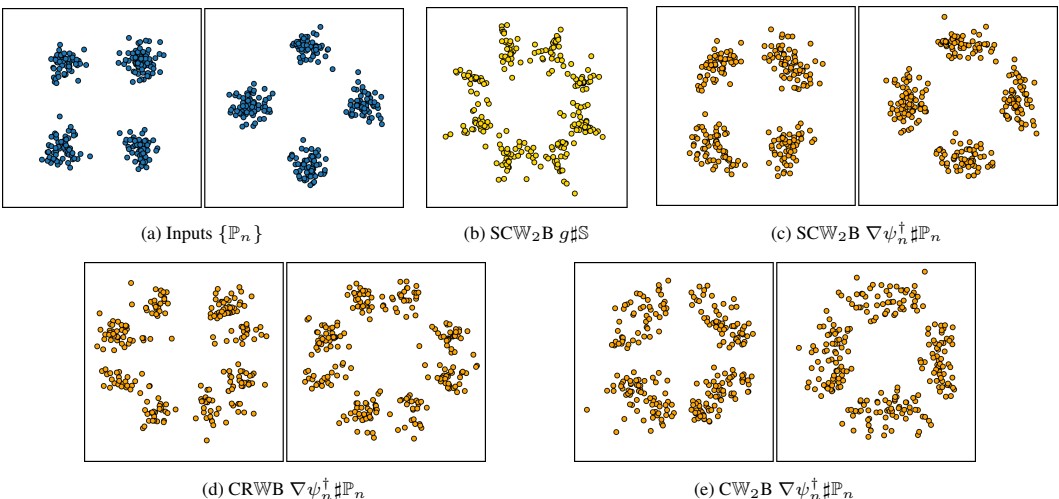

Figure 4: Barycenter of a two 2D Gaussian mixtures.

## C.2 METRICS

The unexplained variance percentage (UVP) (introduced in Section 5) is a natural and straightforward metric to assess the quality of the computed barycenter. However, it is difficult to compute in high dimensions: it requires computation of the Wasserstein-2 distance. Thus, we use different but highly related metrics $\mathcal{L}^2$-UVP and $\mathbb{BW}_2^2$-UVP.

To access the quality of the recovered potentials $\{\psi_n^\dagger\}$ we use $\mathcal{L}^2$-UVP defined in (17). $\mathcal{L}^2$-UVP compares not just pushforward distribution $\nabla\psi_n^\dagger\sharp\mathbb{P}_n$ with the barycenter $\overline{\mathbb{P}}$, but also the resulting transport map with the optimal transport map $\nabla\psi_n^*$. It bounds $\mathrm{UVP}(\nabla\psi_n^\dagger\sharp\mathbb{P}_n)$ from above, thanks to (Korotin et al., 2019a, Lemma A.2). Besides, $\mathcal{L}^2$-UVP naturally admits *unbiased* Monte Carlo estimates using random samples from $\mathbb{P}_n$.

For measure-based optimization method, we also evaluate the quality of the generated measure $g\sharp\mathbb{S}$ using Bures-Wasserstein UVP defined in (18). For measures $\mathbb{P}, \mathbb{Q}$ whose covariance matrices are not degenerate, $\mathbb{BW}_2^2$ is given by

$$\mathbb{BW}_2^2(\mathbb{P},\mathbb{Q}) = \frac{1}{2}\|\mu_\mathbb{P} - \mu_\mathbb{Q}\|^2 + \Big[\frac{1}{2}\operatorname{Tr}\Sigma_\mathbb{P} + \frac{1}{2}\operatorname{Tr}\Sigma_\mathbb{Q} - \operatorname{Tr}(\Sigma_\mathbb{P}^{\frac{1}{2}}\Sigma_\mathbb{Q}\Sigma_\mathbb{P}^{\frac{1}{2}})^{\frac{1}{2}}\Big].$$

Bures-Wasserstein metric compares $\mathbb{P}, \mathbb{Q}$ by considering only their first and second moments. It is known that $\mathbb{BW}_2^2(\mathbb{P},\mathbb{Q})$ is a lower bound for $\mathbb{W}_2^2(\mathbb{P},\mathbb{Q})$, see (Dowson & Landau, 1982). Thus, we have $\mathbb{BW}_2^2$-$\mathrm{UVP}(g\sharp\mathbb{S}) \leq \mathrm{UVP}(g\sharp\mathbb{S})$. In practice, to compute $\mathbb{BW}_2^2$-$\mathrm{UVP}(g\sharp\mathbb{S})$, we estimate means and covariance matrices of distributions by using $10^5$ random samples.

## C.3 CYCLE CONSISTENCY AND CONGRUENCE IN PRACTICE

To assess the effect of the regularization of cycle consistency and the congruence condition in practice, we run the following sanity checks.

For cycle consistency, for each input distribution $\mathbb{P}_n$ we estimate (by drawing samples from $\mathbb{P}_n$) the value $\|\overline{\nabla\psi_n^\ddagger} \circ \nabla\psi_n^\dagger(x) - x\|_{\mathbb{P}_n}^2/\mathrm{Var}(\mathbb{P}_n)$. This metric can be viewed as an analog of the $\mathcal{L}^2$-UVP that we used for assessing the resulting transport maps. In all the experiments, this value does not exceed 2%, which means that cycle consistency and hence conjugacy are satisfied well.

For the congruence condition, we need to check that $\sum_{n=1}^N \alpha_n\psi_n^\dagger(x) = \|x\|^2/2$. However, we do not know any straightforward metric to check this exact condition that is scaled properly by the variance of the distributions. Thus, we propose to use an alternative metric to check a slightly weaker condition on gradients, e.g., that $\sum_{n=1}^N \alpha_n\nabla\psi_n^\dagger(x) = x$. This is weaker due to the ambiguity of the additive constants. For this we can compute $\|\sum_{n=1}^N \alpha_n\nabla\psi_n^\dagger(x) - x\|_{\overline{\mathbb{P}}}^2/\mathrm{Var}(\overline{\mathbb{P}})$, where the denominator is

the variance of the true barycenter. We computed this metric and found that it is also less than 2% in all the cases, which means that congruence condition is mostly satisfied.

## C.4 TRAINING HYPERPARAMETERS

The code is written using the PyTorch framework. The networks are trained on a single GTX 1080Ti.

### C.4.1 WASSERSTEIN-2 CONTINUOUS BARYCENTERS (C$\mathbb{W}_2$B, OUR METHOD)

**Regularization**. We use $\tau = 5$ and $\hat{\mathbb{P}} = \mathcal{N}(0, I_D)$ in our congruence regularizer $\tau \cdot \mathcal{R}_1^{\hat{\mathbb{P}}}$. We use $\lambda = 10$ for the cycle regularization $\lambda \cdot \mathcal{R}_2^{\mathbb{P}^n}$ for all $n = 1, 2, \ldots, N$.

**Neural Networks (Potentials)**. To approximate potentials $\{\psi_n^\dagger, \overline{\psi_n^\ddagger}\}$ in dimension $D$, we use
$$\text{DenseICNN}[2; \max(64, 2D), \max(64, 2D), \max(32, D)]$$
with CELU activation function. DenseICNN is an input-convex dense architecture with additional convex quadratic skip connections. Here $2$ is the rank of each input-quadratic skip-connection's Hessian matrix. Each following number $\max(\cdot, \cdot)$ represents the size of a hidden dense layer in the sequential part of the network. For detailed discussion of the architecture see (Korotin et al., 2019a, Section B.2).

**Training process**. We perform training according to Algorithm 1 of Appendix A. We set batch size $K = 1024$ and balancing coefficient $\gamma = 0.2$. We use Adam optimizer by (Kingma & Ba, 2014) with a fixed learning rate $10^{-3}$. The total number of iterations is set to 50000.

### C.4.2 SCALABLE COMPUTATION OF WASSERSTEIN BARYCENTERS (SC$\mathbb{W}_2$B)

**Generator Neural Network**. For the input noise distribution of the generative model we use $\mathbb{S} = \mathcal{N}(0, I_D)$. For the generative network $g : \mathbb{R}^D \to \mathbb{R}^D$ we use a fully-connected sequential ReLU network with hidden layer sizes
$$[\max(100, 2D), \max(100, 2D), \max(100, 2D)].$$
Before the main optimization, we pre-train the network to satisfy $g(z) \approx z$ for all $z \in \mathbb{R}^D$. This has been empirically verified as a better option than random initialization of network's weights.

**Neural Networks (Potentials)**. We used exactly the same networks as in Subsection C.4.1.

**Training process**. We perform training according to the min-max-min procedure described by (Fan et al., 2020, Algorithm 1). The batch size is set to 1024. We use Adam optimizer by (Kingma & Ba, 2014) with fixed learning rate $10^{-3}$ for potentials and $10^{-4}$ for generative network $g$. The number of iterations of the outer cycle (*min*-max-min) number of iterations is set to 15000. Following (Fan et al., 2020), we use 10 iterations per the middle cycle (min-*max*-min) and 6 iterations per the inner cycle (min-max-*min*).

### C.4.3 CONTINUOUS REGULARIZED WASSERSTEIN BARYCENTERS (CR$\mathbb{W}$B)

**Regularization**. [CR$\mathbb{W}$B] method uses regularization to keep the potentials conjugate. The authors impose entropy or $\mathcal{L}^2$ regularization w.r.t. some proposal measure $\hat{\mathbb{P}}$; see (Li et al., 2020, Section 3) for more details. Following the source code provided by the authors, we use $\mathcal{L}_2$ regularization (empirically shown as a more stable option than entropic regularization). The regularization measure $\hat{\mathbb{P}}$ is set to be the uniform measure on a box containing the support of all the source distributions, estimated by sampling. The regularization parameter $\epsilon$ is set to $10^{-4}$.

**Neural Networks (Potentials)**. To approximate potentials $\{\psi_n^\dagger, \overline{\psi_n^\ddagger}\}$ in dimension $D$, we use fully-connected sequential ReLU neural networks with layer sizes given by
$$[\max(128, 4D), \max(128, 4D), \max(128, 4D)].$$
We have also tried using DenseICNN architecture, but did not experience any performance gain.

**Training process**. We perform training according to (Li et al., 2020, Algorithm 1). We set batch size to 1024. We use Adam optimizer by (Kingma & Ba, 2014) with fixed learning rate $10^{-3}$. The total number of iterations is set to 50000.

