# OpenReview forum: "Continuous Wasserstein-2 Barycenter Estimation without Minimax Optimization"
_ICLR.cc/2021/Conference — ICLR 2021 Poster_

### Official Review · AnonReviewer2 · 2020-10-28
**The paper carefully derives a novel non-adversarial objective via two regularization terms for finding the Wasserstein barycenter maps via convex potential functions.**

**Rating:** 7
**Confidence:** 3

**Review:**

**Summary**
The paper derives the barycenter mapping problem as an optimization over *congruent* convex functions---each convex potential corresponding to a component distribution.  Congruency is a property on the set of optimal potential functions that ties them together.  However, this optimization is quite challenging and so the paper derives an principled objective function that includes two regularization terms.  The first regularization term encourages congruency of the set of convex functions and can be seen as a variational bound on an ideal congruency regularization.  The second regularization term encourages the pairs of convex functions to be conjugate.  The paper proves that the optimal solution of this objective is the true potentials and thus no bias is introduced.  The proposed approach is demonstrated on the tasks of generative modeling (2-256 dimensions), posterior inference, and color pallete barycenters (3D)

**Strengths:**
- Nice problem formulation and setup with respect to prior methods.
- The derivation of the final objective function is clearly laid out and well-motivated.  Each problem that is encountered is explained and then a solution or approximation is introduced.
- The theoretical results give appropriate grounding for the approach.
- The empirical results outperform prior potential-based methods for barycenters.

**Weaknesses:**
- It is unclear if this method can scale in terms of samples and dimensions.  What is the computational cost of estimating these input convex neural networks?  Can you provide (approximate) wall-clock times for the various methods and dimensionalities? What are the key computational bottlenecks either memory-wise or computation-wise?
- The experiments seem small scale with a max dimension of 256.  Barycenters for high dimensional real-world data (e.g., even MNIST (784D)) or some other high-dimensional real-world dataset would improve the paper.
- The paper lacks comparison to methods that do not recover 2N potential functions. What are the closest methods for barycenter that do not use potential functions?  For example, could the algorithms be compared to discretized barycenter algorithms to show the breakdown in higher dimensions?

**Other comments or questions**
- Is $D$ in equation 5 supposed to be $N$?
- Some typos above Eqn. 12.

**Update after author response**

I appreciated the authors response to the scalability and raw computation times.  Thank you also for the additional comparison to a non-potential function method.  This will be a good comparison.  My main concerns were answered, and I still think this is a good paper.

---

> ### Author Response · Authors · 2020-11-15
> **Answer to AnonReviewer2**
>
> Thank you for taking the time to provide constructive comments and critical feedback. We will fix the typos you mentioned and proofread our draft more carefully. Please find below our answers to your questions that do not overlap with those of other reviewers.
>
> **Q: The paper lacks comparison to methods that do not recover 2N potential functions. What are the closest methods for barycenter that do not use potential functions? For example, could the algorithms be compared to discretized barycenter algorithms to show the breakdown in higher dimensions?**
>
> Probably the closest method for computing the barycenters that does not use potential functions is the free-support barycenter computation approach by Cuturi et al. (2014). It is a discrete method that approximates the barycenter with a discrete distribution. The method uses an iterative approach similar to Álvarez-Esteban (2016), but the converged fixed point may not be the true barycenter. As we noted in the general answer to all the reviewers, we plan to add comparison with this discrete method.

---

### Official Review · AnonReviewer3 · 2020-10-28

**Rating:** 7
**Confidence:** 5

**Review:**

Summary:

The paper considers the Wasserstein Barycenter problems in the continuous setting. In particular, the authors propose an algorithm to compute the Wasserstein-2 barycenter when only samples from the marginals are accessible. Some theoretical analysis of this method is presented. Several numerical examples are carried out to compare this method with two other recently proposed methods.

Reasons for score:

The proposed algorithm utilizes an interesting regularization of the dual formulation of Wasserstein-2 Barycenter, resulting in a single minimization problem instead of a min-max problem. This algorithm is properly justified by theoretical results as well as numerical experiments.

Pros:

1. The paper provides theoretical results on the consistency of the proposed algorithm.

2. The experiments are overall good and clear.

2. The paper is well-written and easy to follow.

Cons:

1. High-dimensional examples other than the simple Gaussian setting are missing.

2. There is no analysis of computational complexity of the proposed algorithm. Also, the training expense is not reported.

3. The double gradient in the second regularization term could be expensive to evaluate.

Questions:

1. In both Theorem 4.1 and 4.2, the smoothness of the potentials is crucial. If the smoothness B is too large, then the bound presented is essentially useless. Please comment on it.

---

> ### Author Response · Authors · 2020-11-15
> **Answer to AnonReviewer3**
>
> Thank you for your constructive feedback. Please find below our answer to your questions that do not overlap with those of other reviewers.
>
> **Q: In both Theorem 4.1 and 4.2, the smoothness of the potentials is crucial. If the smoothness $\mathcal{B}$ is too large, then the bound presented is essentially useless. Please comment on it.**
>
> The smoothness coefficient $\mathcal{B}$ of a potential function measures its complexity. Larger $\mathcal{B}$ implies a larger Lipschitz constant for the transport map, which is the gradient of the potential. That is, a large $\mathcal{B}$ leads to the transport map warping the source distribution more drastically. From this perspective, the bound $\mathbb{W}_2^2(\nabla \psi_n\sharp\mathbb{P}_n, \overline{\mathbb{P}}) \le \frac{\mathcal{B}\Delta}{\alpha_n}$ following (10) states that the less complex the fitted potential is, the better theoretical guarantees one may obtain.
>
> As you note, a larger $\mathcal{B}$ (less smooth potentials) indeed gives weaker guarantees. But let us ask the following question: when do we actually need potentials with high $\mathcal{B}$ to approximate the transport map? Such potentials are required only when the true optimal transport map (gradient of the optimal potential) is highly distorted, i.e., has a large Lipschitz constant. For such a difficult problem, guarantees are expected to be weaker. In particular, if the true transport map is unbounded, then it is hard to provide any guarantee. This is supported by the fact that smoothness of optimal transport maps typically improves learning guarantees (see, e.g., https://arxiv.org/pdf/1905.05828.pdf) and, in general, smoothness is a common assumption in optimal transport papers (see https://arxiv.org/abs/1905.10812).
>
> We also emphasize that empirically in low dimensions, when the smoothness constants of the optimal potentials are quite large, our method still captures the barycenter distribution well, as seen in Figure 4b in Appendix.
>
> In practice, it is possible to enforce smoothness of the potentials by clipping the network's weights to a bounded set or by adding an additional penalty term. Korotin et al. (2019) [Section B.1] provides some alternative options to impose strong smoothness via cycle consistency.

---

### Official Review · AnonReviewer4 · 2020-10-28
**Neural based method for continuous Wasserstein-2 barycenters, with good performance**

**Rating:** 6
**Confidence:** 4

**Review:**

This work introduces a new Wasserstein-2 barycenter computation method. The authors first derive the dual formulation of the Wasserstein-2 barycenter problem, and then parametrize the convex potentials by ICNNs. The congruent and conjugacy conditions are enforced by regularization terms, respectively. They then show that the algorithm can find a good barycenter if the objective function is properly minimized.

Pros:
1. The algorithm does not introduce bias.
2. The algorithm does not require minimax, which is efficient.
3. The empirical performance is much better than existing methods, probably due to the above two reasons.

Areas to improve:
1. It is good that the empirical analysis include how the performance change w.r.t. D. It would be better if there is a similar analysis to N. Furthermore, since 2N ICNNs are needed to be trained, it would be better if the training time is also reported, so that we can have a more comprehensive understanding of the method. Will there be a setting that discrete method can be faster than the proposed method to enforce comparable approximation error (say, large N for 3D applications)?
2. Since the congruent and conjugacy conditions are enforced by regularizations, they are not guaranteed to be satisfied. Therefore, it would be better if there is an experiment showing that how the conditions are satisfied.
3. The first section of related work should also briefly include https://arxiv.org/abs/1605.08527 and https://arxiv.org/abs/1905.00158.

After rebuttal:

The additional experiment results provided in the rebuttal stage suggests the efficiency of the proposed method, as well as the congruent and conjugacy conditions are approximately satisfied. I therefore believe this paper should be accepted.

---

> ### Author Response · Authors · 2020-11-15
> **Answer to AnonReviewer4**
>
> Thank you for your insightful feedback. We will add the additional references you suggested. Please find below our answers to your questions that do not overlap with those of other reviewers.
>
> **Q: How well does the proposed algorithm scale with $N$, the number of input distributions?**
>
> We have tested with larger $N$ and empirically our method seems to scale well. Currently, we are preparing experiments to compare the metrics in Table 1 for $N=20$ in dimension $128$, and the results will be available in a few days. Please let us know if this proposed experiment is enough to address the question of scalability with respect to $N$.
>
> **Q: Will there be a setting where a discrete method can be faster than the proposed method, say, for large $N$ in 3D?**
>
> A particularly efficient algorithm for measures supported on a discrete 3D grid is proposed by Solomon (2015), where fast Gaussian convolution is employed to speed up Sinkhorn iterations. However, their algorithm is impractical in higher dimensions, since the number of grid elements will be enormous for a sufficient approximation of the barycenter support. The support of their barycenter is also fixed and depends on the resolution of the discretized grid, making a direct comparison with our method incompatible. Additionally, compared to our setting, their approach contains bias coming from entropic regularization which can also become numerically unstable for small epsilons. The running time for a single iteration of both Solomon (2015) and our method scale linearly in $N$, the number of input distributions.
>
> **Q: Could there be an additional experiment showing how well the congruence and conjugacy conditions are satisfied?**
>
> Following your request, we have computed the following additional numbers.
>
> For cycle consistency, for each input distribution $\mathbb{P}\_n$ we estimate (by drawing samples from $\mathbb{P}\_n$) the value
> $||\nabla \overline{\psi^{\ddagger}\_{n}} \circ \nabla\psi^{\dagger}\_{n}(x) - x||\_{\mathbb{P}\_n}^{2} / \text{Var}(\mathbb{P}\_n)$. This metric can be viewed as an analog of the $\mathcal{L}^2$-UVP that we used for assessing the resulting transport maps. In all the experiments, this value does not exceed 2\%, which means that cycle consistency and hence conjugacy are satisfied well. This is further verified in the 2D case where we visualize the pushforward of samples from $\mathbb{P}\_n$ by $\nabla \overline{\psi^{\ddagger}\_{n}} \circ \nabla\psi^{\dagger}\_{n}$; we obtain the original samples almost perfectly.
>
> For the congruence condition, we need to check that $\sum\_{n=1}^{N}\alpha\_{n}\psi^{\dagger}\_{n}(x)=||x||^2/2$. However, we do not know any straightforward metric to check this exact condition that is scaled properly by the variance of the distributions. Thus, we propose to use an alternative metric to check a slightly weaker condition on gradients, e.g., that $\sum\_{n=1}^{N}\alpha\_{n}\nabla\psi^{\dagger}\_{n}(x)=x$. This is weaker due to the ambiguity of the additive constants. For this we can compute $||\sum\_{n=1}^{N}\alpha\_{n}\nabla\psi^{\dagger}\_{n}(x) - x||\_{\overline{\mathbb{P}}}^{2} / \text{Var}(\overline{\mathbb{P}})$, where the denominator is the variance of the true barycenter. We computed this metric and found that it is also less than 2\% in all the cases, which means that congruence condition is mostly satisfied.
>
> These numbers will be added to the final draft of the paper.

---

### Official Review · AnonReviewer1 · 2020-10-30
**Upper bound for error needs detailed explanation**

**Rating:** 6
**Confidence:** 4

**Review:**

This paper proposes a method to scalably compute Wasserstein-2 barycenters given samples from input measures. In general the authors also allow for continuous measure settings. Inspired by Li et al. (2020) the paper uses a potential-based approach and recovers the barycenter by using gradients of the potentials as pushforward maps.

In general, I feel this paper is well-written and provides a fast solution to a meaningful problem, thereby supporting the claim of novelty. The theoretical developments in the paper are reasonable and the experiments carried out are quite decent, both in simulation and real-data settings.

The only point that bothers me is the approximation used. It would be great if the authors could give an extensive and detailed understanding of settings where the upper bound in Eq.(10) in the main text is small thereby leading to a good approximation.

---

> ### Author Response · Authors · 2020-11-15
> **Answer to AnonReviewer1**
>
> Thank you for your valuable feedback. Please find below our answer to your questions that do not overlap with those of other reviewers.
>
> **Q: Could the authors explain the setting where the upper bound in (10) is small thereby leading to a good approximation?**
>
> Thanks for pointing out an intricate point of Theorem 1 and (10). The purpose of Theorem 1 is to answer the following question: if we approximate the dual problem (8) well-enough by relaxing the congruence constraint, how far are we from the truth barycenter in the primal formulation (compared to the pushforward $(\nabla \psi_n)\sharp \mathbb{P}_n$)? This is an important question from the computational point of view, that is, to our knowledge, not addressed in previous work such as Li et al. (2020), where only strong duality is showed (i.e. at optimality, primal and dual objectives agree).
>
> If the congruence mismatch term in (9) is non-positive, then (10) says the dual gap upper bounds the primal gap. This motivates the congruence regularization term. If the computed dual potentials are precisely the optimal ones, then in (10) the inequalities will become equalities (by strong duality); moreover, if they are $\Delta$ away from the optimal ones, then the primal gap would be bounded by $\frac{\Delta \mathcal{B}}{\alpha_n}$, so the recovered barycenters will not be too far from the ground truth in terms of the Wasserstein distance. Thus, we just need to focus on solving the dual problem, making $\Delta$ as small as possible. An analogous result is given in Theorem 4.2, where we included all the regularization terms but with additional regularity assumptions.
>
> The question of how well we can solve the dual problem depends on the practical aspects: the parameterization of the potentials, the complexity of the truth barycenter, and the optimization procedure. Although we did not directly measure the duality gap, we did compare the difference between (the gradient of) the computed dual potentials and the optimal ones using the $\mathcal{L}^2$-UVP metric (17), and our method has been shown to consistently outperform the alternatives.

---

### Author Response · Authors · 2020-11-15
**Answers To all reviewers**

We thank the reviewers for their valuable feedback and critical comments. Here we address questions raised by multiple reviewers.

---

**Computational expense of training**

Reviewers 2 and 3 asked about the training cost of our algorithm in terms of both time and memory. In all the experiments presented in the paper, our method converges in under 10 minutes. The maximum GPU memory usage at any point was $<2$ GB.

For a single training iteration, the time complexity of both forward (i.e., evaluation) and backward (i.e., computing the gradient with respect to the parameters) passes through the objective function (14) is $O(NT)$. Here $N$ is the number of input distributions and $T$ is the time taken by evaluating each individual potential (parameterized as a neural network) on a batch of points sampled from either $\mathbb{P}_n$ or $\widehat{\mathbb{P}}$.

This claim follows from the well-known fact that gradient evaluation $\nabla_\theta h_\theta(x)$ of $h_\theta: \mathbb{R}^{D} \to \mathbb{R}$, when parameterized as a neural network, requires time proportional to the size of the computational graph. Hence, gradient computation requires computational time proportional to the time for evaluating the function $h_\theta(x)$ itself. The same holds when computing the derivative with respect to $x$. Then, for instance, computing the term $\nabla\overline{\psi_{n}^{\ddagger}}\circ \nabla\psi_{n}^{\dagger} (x)$ in (14) takes $O(T)$ time. The gradient of this term with respect to $\theta$ also takes $O(T)$ time: Hessian-vector products that appear can be calculated in $O(T)$ time using the famous Hessian trick (Pearlmutter 1993).

In practice, we compute all the gradients using PyTorch's automatic differentiation on a single GTX 1080Ti GPU. We empirically measured that for our DenseICNN potentials, the computation of their gradient w.r.t. input $x$, i.e., $\nabla \psi^\dagger(x)$, requires roughly 3-4x more time than the computation of $\psi^\dagger(x)$.

The memory complexity of automatic differentiation is more complicated and depends on the automatic differentiation tool. Empirically, we did not experience any difficulty with memory usage when computing the barycenter of 4 distributions in dimension 256 with a batch size of 1024.

---

**Non-Gaussian high-dimensional experiments and discrete methods**

Reviewers 2 and 3 asked about the performance of our algorithm in non-Gaussian high-dimensional experiments.

Please note that we have included in Section 5.1 experiments with location-scatter families in dimensions up to 256, and the Gaussian case is just one example of such a family. The tricky part of conducting barycenter experiments is that the ground truth is often not known. The location-scatter families are probably the most complicated example of continuous distributions for which the ground truth barycenter can be computed using the iterative algorithm from Álvarez-Esteban (2016). Another scenario where we can have a grasp of the ground truth barycenter is in subset posterior aggregation (Section 5.2), in which the barycenter of subset posteriors converges to the true posterior. However, Bayesian inference on high-dimensional parameter space is a challenging problem on its own.

We comment that the previous works on continuous barycenters, e.g., Li et al. (2020) and Fan et al. (2020), considered only the Gaussian case in much smaller dimensions. Li et al. (2020) tested their algorithm only in dimensions up to 8. The highest dimension (256) we considered is very large for continuous optimal transport in general. Regarding barycenters, the majority of the papers (especially those based on discrete methods) consider only small dimensions such as 2 or 3.

In the original draft, we did not compare with discrete methods, because Li et al. (2020) [Table 1] and Fan et al. (2020) [Figure 6b] have already shown that discrete methods fail drastically as the dimension increases. Following your request, we are currently running experiments on the discrete method by Cuturi et al. (2014) to compare against our method. The results will be provided in a few days.

---

Please let us know whether the aforementioned additional experiments will adequately address your questions about time/memory complexity and comparison with the discrete approaches.  If not, we are happy to run additional experiments at your request during the rebuttal period.

---

> ### Public Comment · ~Pavel_Dvurechensky1 · 2020-11-16
> **Comparison with the semi-discrete approach for WB**
>
> For your information: besides the discrete-discrete method by Cuturi & Doucet (2014) there are algorithms for the semi-discrete setting, in which only the barycenter is discretized:
> http://papers.neurips.cc/paper/6858-parallel-streaming-wasserstein-barycenters
> https://proceedings.neurips.cc/paper/2018/hash/161882dd2d19c716819081aee2c08b98-Abstract.html
> Both algorithms are distributed, which helps to scale up computations. Also the first algorithm does not rely on entropic regularization.

---

> > ### Author Response · Authors · 2020-11-17
> > **Response to comment**
> >
> > Thank you for pointing out the references and we will include them in the related-work section of the revision.
> >
> > We are aware of these two methods but did not compare to them in experiments because they use a discrete fixed support for the barycenter, which will not work in higher dimensions. For example, Li et al. (2020) has compared to Staib (2017) for the Gaussian case in $\mathbb{R}^4$ with $10^5$ support points that resulted in significantly worse numbers compared to the free-support methods - this is reported on page 8 of Li et al. (2020).

---

> ### Author Response · Authors · 2020-11-18
> **Rebuttal Revision**
>
> Dear reviewers, we have revised the paper according to your comments. Please consider the updated submission.
>
> The edits are highlighted by the blue color in the revised version of the paper. The main edits are listed below.
>
> **(R2, R3)** We added the comparison (section 5.1) of our method with the method by Cuturi and Doucet (2014).
>
> **(R2, R3)** We added the discussion of computational complexity of a single gradient step of or method to Appendix A of the paper.
>
> **(R4)** We enhanced the related work section according to reviewer's recommendation.
>
> **(R4)** To demonstrate the scalability of our method with a large number of input distributions $N$, we added the experiments with the $N=20$ to the end of Section 5.1.
>
> **(R4)** We added Appendix C.3 in which we explain that the cycle-consistency and congruence conditions actually satisfied in our model in practice.
>
> If there are any additional changes you suppose we should perform, please kindly suggest.

---

### Decision · Program_Chairs · 2021-01-07
**Final Decision**

**Decision:**

Accept (Poster)

**Comment:**

The authors propose the 2-Wasserstein barycenter problem between measures. The authors propose a novel formulation that leverages a condition (congruence) that the optimal transport (Monge) maps, here parameterized as potentials, must obey at optimality. The introduce various regularizers to encourage that property. The idea is demonstrated on convincing synthetic experiments and on a simple color transfer problem. Although experiments are a bit limited, I do believe, and follow here the opinion of all reviewers, that there is novelty in this approach, and that this paper is a worthy addition to the recent line of work trying to leverage ICNNs/Brenier's theorem to solve OT problems.